# Autonomic ganglionic injection of α-synuclein fibrils as a model of pure autonomic failure α-synucleinopathy

Xue-Jing Wang[1,2,10 ✉], Ming-Ming Ma[3,10], Le-Bo Zhou[1,2], Xiao-Yi Jiang[1,2], Miao-Miao Hao[1,2], Robert K.F. Teng[4], Erxi Wu [5], Bei-Sha Tang[6,7 ✉], Jia-Yi Li[8,9 ✉], Jun-Fang Teng [1,2 ✉] & Xue-Bing Ding[1,2,7 ✉]

α-Synucleinopathies are characterized by autonomic dysfunction and motor impairments. In the pure autonomic failure (PAF), α-synuclein (α-Syn) pathology is confined within the autonomic nervous system with no motor features, but mouse models recapitulating PAF without motor dysfunction are lacking. Here, we show that in TgM83[+/−] mice, inoculation of α-Syn preformed fibrils (PFFs) into the stellate and celiac ganglia induces spreading of α-Syn pathology only through the autonomic pathway to both the central nervous system (CNS) and the autonomic innervation of peripheral organs bidirectionally. In parallel, the mice develop autonomic dysfunction, featured by orthostatic hypotension, constipation, hypohidrosis and hyposmia, without motor dysfunction. Thus, we have generated a mouse model of pure autonomic dysfunction caused by α-Syn pathology. This model may help define the mechanistic link between transmission of pathological α-Syn and the cardinal features of autonomic dysfunction in α-synucleinopathy.

[1] Department of Neurology, The First Affiliated Hospital of Zhengzhou University, Zhengzhou, Henan 450052, China. [2] Institute of Parkinson and Movement Disorder, Zhengzhou University, Zhengzhou, Henan 450052, China. [3] Department of Neurology, Affiliated People's Hospital of Zhengzhou University, Henan Provincial People's Hospital, Zhengzhou, Henan 450003, China. [4] Collage of Electronic and Information Engineering, Shenzhen University, Shen Zhen, Guangdong 518060, China. [5] Neuroscience Institute and Department of Neurosurgery, Baylor Scott & White Health, Temple, Texas 76508, USA. [6] National Clinical Research Center for Geriatric Disorders, Xiangya Hospital, Changsha, Hunan 410008, China. [7] Center for Medical Genetics, School of Life Sciences, Central South University, Changsha, Hunan 410008, China. [8] Neural Plasticity and Repair Unit, Wallenberg Neuroscience Center, Department of Experimental Medical Science, Lund University, BMC A10, 221 84 Lund, Sweden. [9] Institute of Health Sciences, China Medical University, 110112 Shenyang, China. [10]These authors contributed equally: Xue-Jing Wang, Ming-Ming Ma. ✉email: fccwangxj2@zzu.edu.cn; tangbeisha@sklmg.edu.cn; lijiayi@cmu.edu.cn; 13838210077@163.com; fccdingxb@zzu.edu.cn

α-Synucleinopathies are a spectrum of progressive neurodegenerative disorders characterized by accumulation of α-synuclein (α-Syn) aggregates in selective regions within the nervous system, including Parkinson's disease (PD), dementia with Lewy bodies (DLB), multiple system atrophy (MSA), and pure autonomic failure (PAF)[1–5]. Different phenotypes in these disorders result from distinct patterns of abnormal α-Syn accumulation in the nervous system. Autonomic dysfunction is common and well known in α-synucleinopathies, appearing more frequently than motor impairments in the early stage of disease[4,6,7]. Among the α-synucleinopathies, PAF is a relatively special type, characterized by slowly progressive autonomic failure without motor dysfunctions[8]. Postmortem histopathological studies demonstrate that Lewy bodies have been identified in the autonomic pathway, including the sympathetic ganglia, intermediolateral nucleus (IML) of the spinal cord, dorsal motor nucleus of vagus (10N), locus coeruleus (LC), substantia nigra, raphe nucleus (RN), and brainstem reticular formation[9–15].

Previous studies support that cell-to-cell templated propagation of α-Syn in a prion-like manner could be induced by exogenous pathological α-Syn in cultured cells and mouse models[16–26]. Accordingly, various mouse models of α-synucleinopathy induced by exogenous inoculation of pathological α-Syn develop α-Syn pathology in different distribution patterns and clinical phenotypes predominantly related to motor dysfunction; however, autonomic dysfunction rarely occurs or is observed in these models. Similarly, PD-associated mutant α-Syn transgenic mice spontaneously develop dyskinesia and urinary dysfunction, without other autonomic dysfunction[27]. To date, the mechanisms underlying autonomic dysfunction in α-synucleinopathies are still largely unknown. There is a lack of appropriate experimental models to fully recapitulate the entire autonomic dysfunction of α-synucleinopathy.

In the present study, we report a model that mimics autonomic dysfunction in different α-synucleinopathies. We demonstrate that inoculation of αSyn preformed fibrils (PFFs) into the stellate ganglia and celiac ganglia of TgM83[+/−] mice can induce transmission of pathological α-Syn through the autonomic pathway to both the CNS and autonomic innervation of peripheral organs bidirectionally. Additionally, αSyn PFFs-inoculated TgM83[+/−] mice exhibited pure autonomic dysfunction including orthostatic hypotension, constipation, and hypohidrosis or anhidrosis, accompanied with olfactory dysfunction. The physical weakness induced by autonomic dysfunction results in movement reduction and inactivity of these mice immediately before their death, but they showed no signs of motor dysfunction (e.g. ataxia, dysmetria, or paralysis) during the disease course. Taken together, we have established a mouse model of pure autonomic dysfunction induced by pathological α-Syn, which will provide a valuable avenue for interrogating progression mechanisms of, and developing disease-modifying therapies for, autonomic dysfunction in αsynucleinopathy.

## Results

**Retrograde transsynaptic spread of pathological α-Syn in the central autonomic pathways**. To investigate whether inoculation of α-Syn PFFs in sympathetic ganglia can transmit α-Syn pathology in vivo, we injected α-Syn PFFs (47.8 ± 23.7 nm, Supplementary Fig. 1c, d, f) into both bilateral stellate ganglia and celiac ganglia of 2-month-old TgM83[+/−] mice (α-Syn PFFs mice). These mice were sacrificed at every month during the 7 months after the injection. Six sections from various locations of central autonomic pathways, such as the 3rd thoracic spinal cord (T3), the 11th thoracic spinal cord (T11), medulla, pons

(lower), pons (upper) and midbrain, were subjected to immunohistochemistry (IHC), immunofluorescence (IF), and Western blot to detect pathological α-Syn.

We detected α-Syn-positive profiles in various levels of central autonomic pathways in α-Syn PFFs mice as early as 2 months post-injection (mpi). Autonomic function-associated neuronal populations exhibited increasingly remarkable formation of α-Syn inclusions. IML and Lamina VII (7Sp) of the thoracic spinal cord showed abundant phosphorylated α-Syn at serine 129 (pα-Syn) positive profiles in α-Syn PFFs mice (Fig. 1a, b). Control centers in the brain regulating cardiac, gastrointestinal, and urinary function developed α-Syn pathology including the arcuate hypothalamic nucleus (Arc), periaqueductal gray (PAG), raphe nucleus (RN), locus coeruleus (LC), Barrington's nucleus (Bar), dorsal nucleus of vagus nerve (10N), ambiguus nucleus (Amb), nucleus of the solitary tract (Sol), the pontine reticular nucleus, oral part (PnO) and caudal part (PnC), the parvicellular reticular nuclei (PCRt) and the gigantocellular reticular nuclei (Gi) (Fig. 1). Pathological α-Syn deposits were undetectable in phosphate-buffered saline (PBS)-injected mice (PBS mice). In order to verify the aggregation of pathological α-Syn in different types of cells in the brain, we performed a series of double IF stainings with α-Syn antibodies and specific cellular marker antibodies. We used tyrosine hydroxylase (TH), choline acetyltransferase (ChAT), tryptophan hydroxylase (TPH) and microtubule-associated protein-2 (MAP-2) for neurons; glial fibrillary acidic protein (GFAP) for astrocytes; and myelin basic protein (MBP) for oligodendrocytes; as well as ubiquitin (Ub) for inclusions (Fig. 2). Double IF results revealed the co-localizations of pα-Syn inclusions with TH in A6 cell group of LC (Fig. 2a) and C1 cell group of medulla oblongata (Fig. 2b), ChAT in Amb (Fig. 2c) and 10N (Fig. 2d), TPH in dorsal raphe nucleus, dorsal part (DRD) (Fig. 2e), MAP-2 in parvicellular reticular nucleus, alpha part (PCRtA) (Fig. 2g). Neurons contain the most abundant pα-Syn-positive inclusions, while no appreciable co-localization of pα-Syn with GFAP (Fig. 2f) or MBP (Fig. 2h) in the pons or the medulla oblongata were observed. Pα-Syn inclusions are positive for Ub in the medulla oblongata (Fig. 2i) and pons (Fig. 2j). We also analyzed the presence of insoluble α-Syn in the stellate ganglia, celiac ganglia, thoracic spinal cord, hypothalamus, and brainstem tissues of α-Syn PFFs mice and PBS mice at 4 mpi by Western blot to further demonstrate α-Syn pathology. More insoluble and pα-Syn were detected in the extracts of α-Syn PFFs mice whereas no phospho-α-Syn was detected in the brain extracts of PBS-injected mice (Fig. 2k–p). Nevertheless, the pathological α-Syn was undetectable in the control center of olfaction, the olfactory bulb (Ob). TH immunostaining of Ob showed that dopaminergic neurons were significantly decreased in α-Syn PFFs mice. In order to further investigate the spreading pattern of α-Syn-positive inclusions in the early stages of pathological processes in TgM83[+/−] mice, unilateral injections were performed in both the right stellate ganglion and the right celiac ganglion. At about 2 mpi, the unilateral α-Syn PFFs-injected mice showed more abundant pα-Syn-positive profiles in the ipsilateral (right) side of the brain and spinal cord (Supplementary Fig. 2g–n). These pα-Syn-positive profiles were mainly distributed in the 7Sp and IML of the right spinal cord (Supplementary Fig. 2g, k). In the brain, the reticular structure was first invaded by pα-Syn PFFs spreading, then, the Gi in the medulla oblongata, the pontine reticular nucleus, PnC and PnO in the pons (Supplementary Fig. 2h–j, l–n). Furthermore, the Bar and PAG in the pons exhibited pα-Syn-positive profiles at the early stages of pα-Syn PFFs injection (Supplementary Fig. 2i, j, m, n). Interestingly, by 3 mpi, pα-Syn-positive profiles appeared in both sides of the brain and spinal cord, thereafter, side dominancy of α-Syn distribution was no longer obvious.

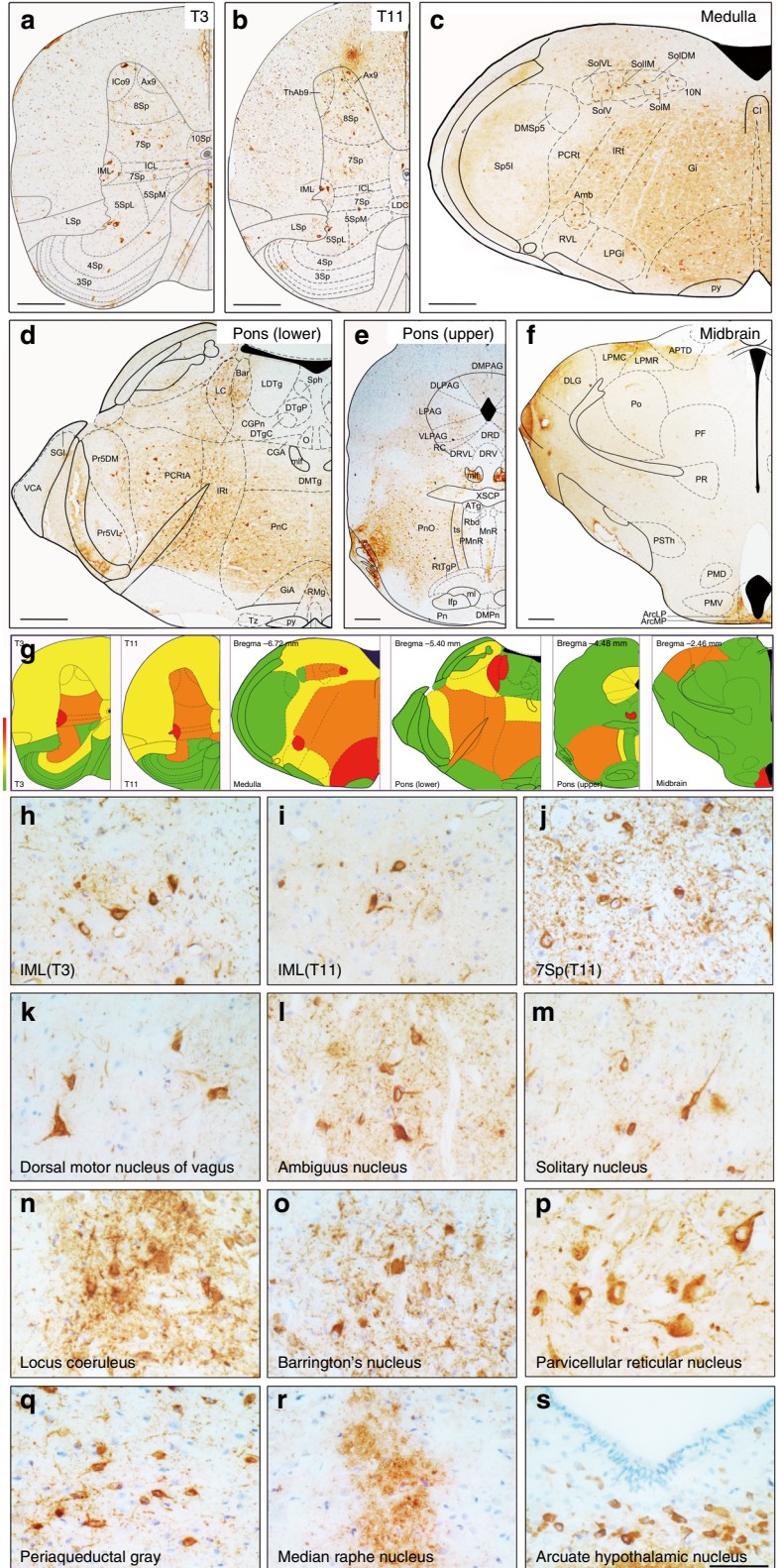

**Fig. 1 Phosphorylated α-Syn pathology in CNS segments of diseased α-Syn PFFs mice. a–f** Representative immunohistochemical results displaying the distribution of pα-Syn pathology in the 3rd thoracic spinal cord (T3) (**a**), the 11th thoracic spinal cord (T11) (**b**), medulla (**c**), pons (lower) (**d**), pons (upper) (**e**) and midbrain (**f**). **g** Schematic displaying the distribution pattern of pα-Syn pathology in diseased α-Syn PFFs mice. Green, yellow, orange, and red represent the severity of α-Syn pathology, ranging from low densities to high densities of inclusions. **h–s** High-magnification immunohistochemical images in different CNS regions from diseased α-Syn PFFs mice show the pα-Syn inclusions in IML of T3 (**h**) and T11 (**i**), 7Sp of T11 (**j**), dorsal motor nucleus of vagus (**k**), ambiguus nucleus (**l**), solitary nucleus (**m**), locus coeruleus (**n**), Barrington's nucleus (**o**), parvicellular reticular nucleus (**p**), periaqueductal gray (**q**), median raphe nucleus (**r**), and arcuate hypothalamic nucleus (**s**). [Scale bars, 500 μm (**a–d**); 1 mm (**e**, **f**); 66 μm (**h–s**)].

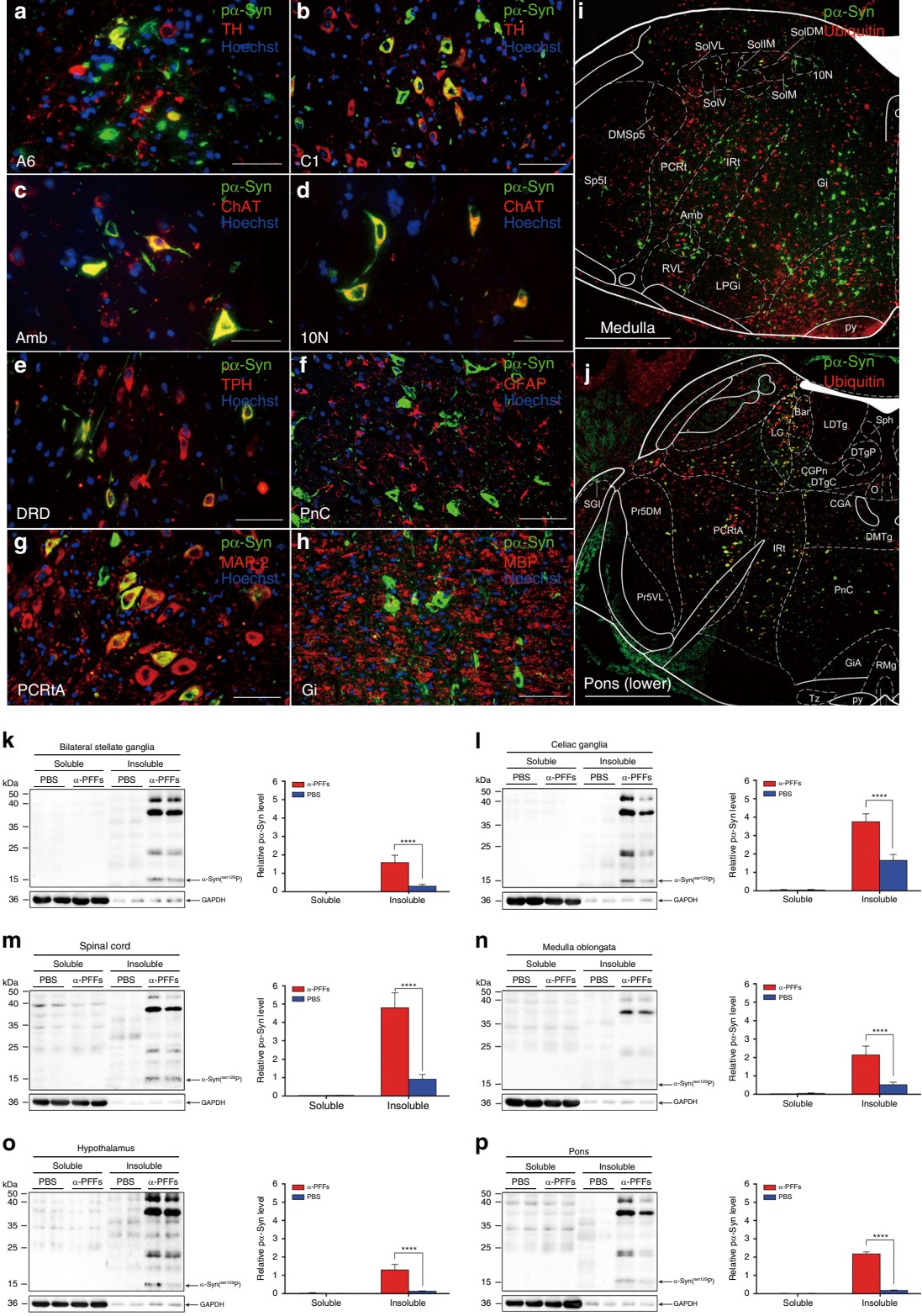

**Anterograde axonal spread of pathological α-Syn to autonomic effector organs**. At 1 mpi, we detected scattered pα-Syn-positive neuronal bodies and axonal profiles in the stellate ganglia and celiac ganglia. To further investigate the spreading pattern of α-Syn pathology in autonomic effector organs, we detected pathological α-Syn deposits in multiple organs of the α-Syn PFFs mice. Pα-Syn pathology in the stellate and celiac ganglia was sparse at 1 mpi, and became more robust at 3 mpi in α-Syn PFFs mice

compared to the PBS-injected mice (Fig. 3a, b, k, l). Autonomic effector organs showed a few scattered pα-Syn inclusions at 2 mpi, including the myocardium (Fig. 3c–f), blood vessels (Fig. 3g, h), gastrointestinal tract (Fig. 3m–r) and skin (Fig. 3i, j, s, t). Over time, pα-Syn inclusions became more condensed and intense. Pα-Syn-positive structures were detected in the myocardial nerve terminals (Fig. 3c, e). Pα-Synimmunoreactive nerve fibers in the cardiac blood vessels were detected in α-Syn PFFs

**Fig. 2 Double immunofluorescence and Western blot analyses of diseased α-Syn PFFs mice. a–j** Double immunofluorescence analysis of pons and medulla oblongata from α-Syn PFFs mice for pα-Syn (green) with TH in A6 cell group of locus coeruleus (red, **a**) and C1 cell group of medulla oblongata (red, **b**), ChAT in ambiguus nucleus (red, **c**) and dorsal motor nucleus of vagus (red, **d**), TPH in dorsal raphe nucleus, dorsal part (red, **e**), GFAP in pontine reticular nucleus, caudal part (red, **f**), MAP-2 in parvicellular reticular nucleus, alpha part (red, g), MBP in gigantocellular reticular nucleus (red, **h**), ubiquitin in medulla (red, **i**) and pons (red, **j**). Co-immunolabeling is represented by signal in yellow. Cell nuclei were counterstained with Hoechst33258 (blue, **a-h**). [Scale bars, 40 μm (**a**, **c**, **d**, **e**, **g**); 50 μm (**b**, **f**, **h**); 500 μm (**i**, **j**)]. (**k–p**) Representative immunoblots of α-Syn in the soluble and insoluble fractions of bilateral stellate ganglia (**k**), celiac ganglia (**l**), spinal cord (**m**), medulla oblongata (**n**), hypothalamus (**o**), and pons (**p**) using the α-Syn (Ser129P) antibody (left of each panel). Blots were probed for GAPDH as a loading control (bottom). Molecular weight markers of migrated protein standards are expressed in kDa. Quantification of soluble and insoluble α-Syn levels in bilateral stellate ganglia, celiac ganglia, spinal cord, medulla oblongata, pons, and hypothalamus (right of each panel, $n = 3$ animals per group). The error bar in panels k-p represents the standard deviation (SD). Data are the means ± SD. Statistical significance was analyzed using the Student's $t$ test and Mann–Whitney test, *$P < 0.05$; n.s., non-significant. Source data is available as a Source Data file.

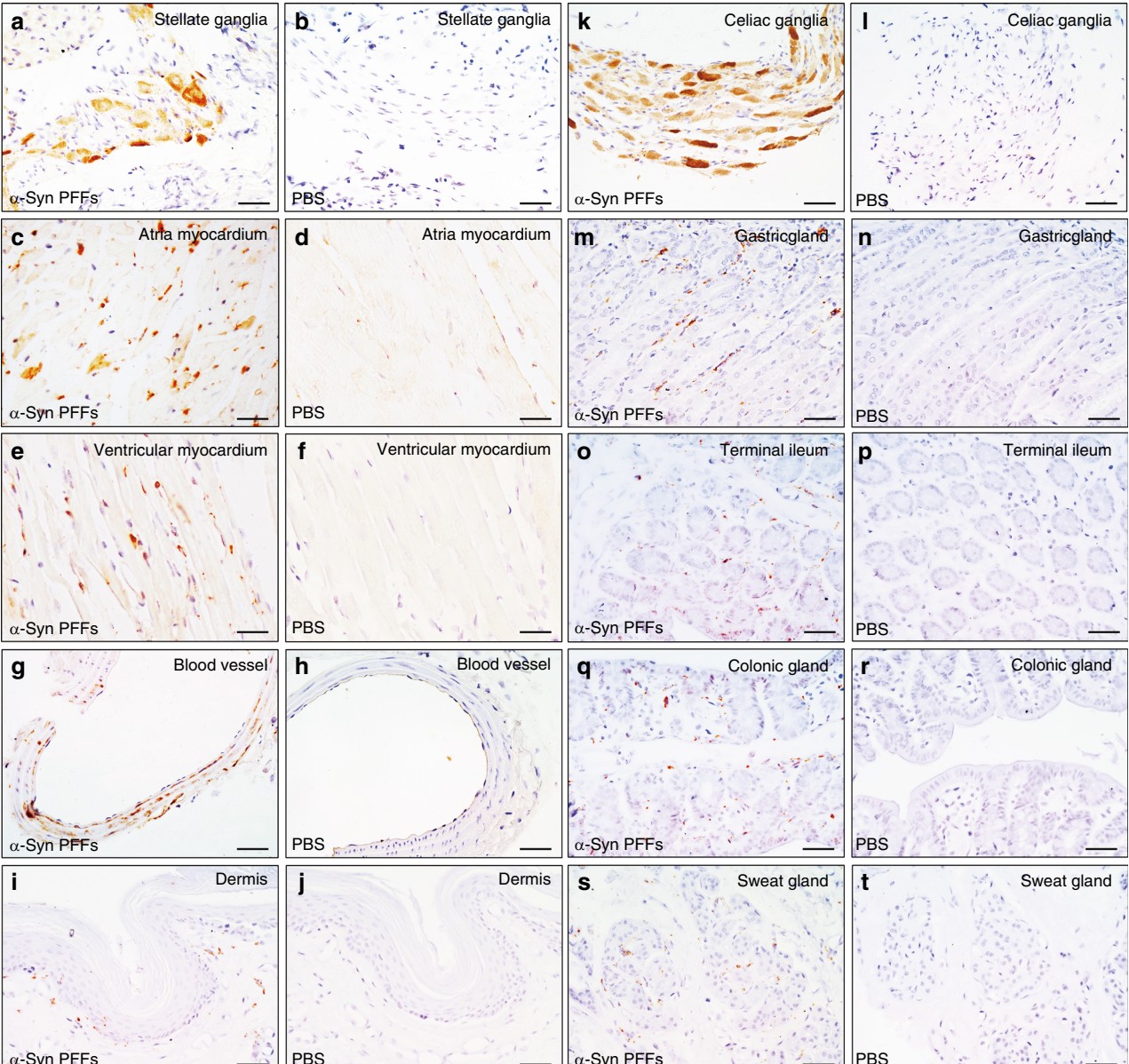

**Fig. 3 Representative immunohistochemical results of multiple peripheral regions. a–t** Pathological α-Syn stained with anti-phospho-α-Syn (Ser 129) antibody. The immunohistochemical images display the distribution of pα-Syn in the stellate ganglia, atria myocardium, ventricular myocardium, blood vessel, dermis, celiac ganglia, gastric gland, terminal ileum, colonic gland, and sweat gland in diseased α-Syn PFFs mice (**a**, **c**, **e**, **g**, **i**, **k**, **m**, **o**, **q**, **s**), but not in PBS mice (**b**, **d**, **f**, **h**, **j**, **l**, **n**, **p**, **r**, **t**). [Scale bars, 50 μm].

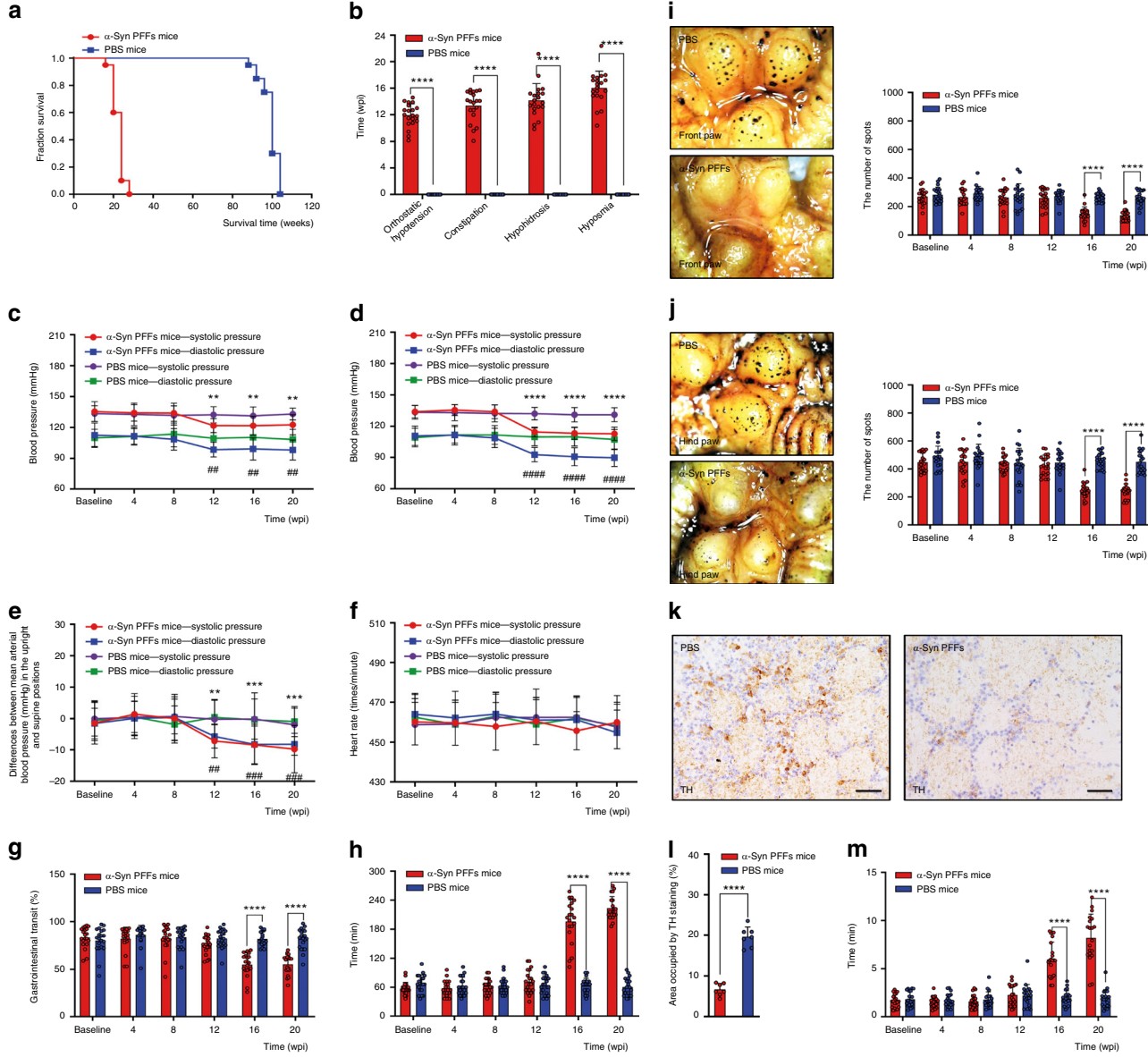

**Fig. 4 Analysis of autonomic function and olfaction in α-Syn PFFs mice and PBS mice. a** Kaplan–Meier survival plot shows survival time (weeks of age) of diseased α-Syn PFFs mice and PBS mice. **b** Onset time (weeks post-injection (wpi)) of orthostatic hypotension, constipation, hypohidrosis, and hyposmia in α-Syn PFFs and PBS mice. **c-h** The average systolic and diastolic blood pressure in the supine position (**c**), the average systolic and diastolic blood pressure in the upright position (**d**), the differences of the mean arterial blood pressure between the upright position and lying position (**e**), the heart rate (**f**), the gastrointestinal transit (**g**), and defecation time of black feces (**h**) of diseased α-Syn PFFs mice and PBS mice over time. **i, j** Sweating test in front (**i**) and hind (**j**) paw of diseased α-Syn PFFs mice and PBS mice over time. **k–m** Olfaction evaluation. Tyrosine hydroxylase (TH) staining in olfactory bulb (Ob) (**k**) of PBS (left) and diseased α-Syn PFFs (right) mice. [Scale bars, 50 μm]. Bar graphs with scattered dots show the percentage of area occupied by TH immunostaining (**l**) and the comparison of the buried food test between diseased α-Syn PFFs mice and PBS mice over time (**m**). $n = 20$ animals per group. The error bar in panels **b-j**, **l**, **m** represents the standard deviation (SD). Data are the means ± SD. Statistical analysis was performed using the Student's $t$ test. **$p < 0.01$, ##$p < 0.01$, ***$p < 0.001$, ###$p < 0.001$, ****$p < 0.0001$, ####$p < 0.0001$. Source data is available as a Source Data file.

mice (Fig. 3g). In contrast to the other areas in the gastrointestinal tissue, the gastric glands, terminal ileum, and colonic glands exhibited abundant punctate or filamentous pα-Syn immunoreactivity arranged in the lamina propria (Fig. 3m, o, q). In the hindlimb skin of α-Syn PFFs mice, pα-Syn immunoreactive fibers were mainly distributed in the dermises and sweat glands (Fig. 3i, s). In unilaterally α-Syn PFFs-injected mice, pα-Syn inclusions were detected in autonomic effector organs at about 3 mpi. The median survival times of α-Syn PFFs mice and PBS mice injected bilaterally are 24 weeks and 100 weeks of age respectively (Fig. 4a). The median survival times of α-Syn PFFs

mice and PBS mice injected unilaterally are 28 weeks and 100 weeks of age, respectively (Supplementary Fig. 2a).

**Orthostatic hypotension.** In addition to the morphological evidence of α-Syn PFFs-induced pathology, we also investigated whether the α-Syn PFFs mice exhibited any autonomic dysfunction, such as orthostatic hypotension, constipation or hypohidrosis, which associates with the pathological α-Syn deposition in the autonomic nerve terminals of the above-mentioned autonomic effector organs. To evaluate the cardiovascular function, we used a noninvasive tail-cuff BP analyzer (CODA, Kent

Scientific) to monitor the arterial blood pressure and heart rate in the supine (0°) position and upright (90°) position. There were no significant differences in the blood pressure or heart rate in the supine and upright positions of the two groups in the first two months after α-Syn injection. Also, no statistically significant differences of the systolic blood pressure (SBP) and diastolic blood pressure (DBP) were detected between the two groups (Fig. 4c, d). At 3 mpi, the levels of SBP and DBP in the upright position were significantly decreased in α-Syn PFFs mice compared with PBS ones, and then maintained at low levels until the end of the experiment (i.e. 20 weeks) (Fig. 4d). Similarly, at the same time, the levels of SBP and DBP in the supine position were slightly decreased in α-Syn PFFs mice (Fig. 4c). Furthermore, a statistically significant reduction in the SBP was detected from supine to upright positions in α-Syn PFFs mice compared with PBS ones, and the reduction in DBP from supine to upright positions also occurred at 3 mpi (Fig. 4e). However, no significant difference in heart rate was observed in the supine and upright positions of the two groups (Fig. 4f). The unilateral α-Syn PFFs-injected mice developed orthostatic hypotension at 14 weeks post-injection (wpi), a 2-week delay compared to the bilaterally injected α-Syn PFFs mice (Fig. 4b, Supplementary Fig. 2b). These data indicate that intra-stellate ganglionic injection of a-syn PFFs results in orthostatic hypotension in TgM83$^{+/-}$ mice (Fig. 4c–f). Taken together, these findings show that inoculation of α-Syn PFFs in stellate ganglia induced the cardiovascular dysfunction.

**Constipation.** In order to assess gastrointestinal function, we determined the gastrointestinal transit and the time. Firstly, the effects of intra-celiac ganglionic injection of α-Syn PFFs on intestinal motility were determined by gastrointestinal transit in mice 16 h after activated carbon administration (0.2 ml/mouse, 10% activated carbon). As shown in Fig. 4g, there is no statistically significant difference between the mean GI transits of each group in the first three months after modeling. At 3.5 mpi, the mean GI transit in the α-Syn PFFs group was 65.1%, which was slower than that of the PBS group (80.1%). Then, the mean GI transit in the α-Syn PFFs group continued to decrease to 55.3% at 5 mpi. Incubation of α-Syn PFFs in the celiac ganglia decreases the GI transit compared with the control, increases constipation and decreases the functional effect. The data described above indicate that intra-celiac ganglionic injection of α-Syn PFFs inhibited the activated carbon passing through the small intestine. In the first three months after modeling, the time to the first black stool defecation (defecation time) in the two groups showed no difference. Then, the defecation time in α-Syn PFFs mice was significantly prolonged compared to PBS mice at 3.5 mpi, and elongated gradually to 5 mpi (Fig. 4h). According to the defecation time, intra-celiac ganglionic injection of α-Syn PFFs resulted in constipation, and some α-Syn PFFs mice became severely constipated at the end of the experiment, while there was no significant abnormality in the GI transit of PBS mice. The unilaterally α-Syn PFFs-injected mice developed constipation at about 16 wpi, which was much delayed compared with the bilaterally injected mice (Fig. 4b, Supplementary Fig. 2b). All of the above observations indicate gastrointestinal dysfunction in α-Syn PFFs mice with progressive aggravation.

**Hypohidrosis in both forepaws and hindpaws.** We utilized the pilocarpine-induced sweat and starch-iodine assay to examine the sweating function of mice. When pilocarpine was subcutaneously injected into the paws of mice, individual sweat glands (represented by black dots) appeared within 5 min, and the number increased in a time-dependent manner (Fig. 4i, j). In α-Syn PFFs mice, the increase in sweat gland number was significantly

attenuated (198.6 ± 74.9 dots per paw at 10 min, $n = 20$; Fig. 4i, j) compared with PBS mice at 3.5 mpi. In addition, the size of each black dot (presumably representing the sweat volume from a single gland) was sharply decreased in α-Syn PFFs mice versus PBS mice. Comparing forepaws with hindpaws in α-Syn PFFs mice, the sweat glands in forepaws were fewer than that in hindpaws. Furthermore, the spots in both forepaws and hindpaws of α-Syn PFFs mice at 5 mpi were fewer than before. The α-Syn PFFs mice injected unilaterally developed hypohidrosis at around 18 wpi, 4 weeks more delayed than that of α-Syn PFFs mice injected bilaterally (Fig. 4b, Supplementary Fig. 2b). Conclusively, these findings reveal that intra-sympathetic ganglionic injection of α-Syn PFFs results in reduced sweat production in α-Syn PFFs mice.

**Hyposmia.** To examine olfactory function, we used the buried food test to evaluate olfactory deficits in mice. The buried food test measures how quickly an overnight-fasted animal can find a small piece of familiar palatable food[28]. As shown in Fig. 4, we compared the latency to locate the small piece of familiar palatable food. α-Syn PFFs mice took significantly longer to locate the buried food compared with PBS mice until 4 mpi (Fig. 4m). Moreover, at 5 mpi some α-Syn PFFs mice could not find cookies even when the cookies were on the surface of the bedding. However, α-Syn PFFs and PBS mice showed similar interest, mobility and attention to the stimulus. The α-Syn PFFs mice injected unilaterally developed hyposmia at 18 wpi, 2 weeks later than that of α-Syn PFFs mice injected bilaterally (Fig. 4b, Supplementary Fig. 2b). Since there were no pα-Syn-positive inclusions detected in the olfactory pathway, including the olfactory cortex and Ob, we tried to find a reason for the olfactory deficits in α-Syn PFFs mice at 4 mpi. Immunohistochemical analysis showed that there was a significant decrease in TH-positive cells and nerves in the Ob compared with PBS mice (Fig. 4k, l), which may induce the olfactory deficits in α-Syn PFFs mice.

**No motor dysfunctions.** As mentioned above, α-Syn PFFs mice developed various autonomic dysfunctions at 3 mpi, however, they did not display any motor dysfunctions. Then, α-Syn PFFs mice gradually became weak and inactive with reduced movements, whereas their posture, gait, muscle strength, motor accuracy and coordination ability were normal as evaluated by behavioral tests. As shown in Fig. 5, the footprint test and hanging wire test indicated that the gait and muscle strength of α-Syn PFFs mice were comparable to that of PBS mice in different motor behavioral tests (Fig. 5d–h). In the beam walking test, α-Syn PFFs mice were observed to be in the inactive state, and to have a significant increase in the duration to cross the beam and the number of errors for its inactive state (Fig. 5a, b), compared with PBS mice. However, animals did not exhibit hindlimb weakness, akinesia, postural instability or action tremor. The rotarod test revealed that reduced movements in α-Syn PFFs mice emerged at 4 mpi, while the PBS mice showed no movement reduction (Fig. 5c). These reflected that the activity of α-Syn PFFs mice was worse in comparison with PBS mice. The slight motor impairments observed in α-Syn PFFs mice may be associated with autonomic dysfunctions, such as cardiovascular and gastrointestinal problems, which induced the fragile and weak state of α-Syn PFFs mice. Therefore, we conclude that α-Syn PFFs mice developed autonomic dysfunction without motor dysfunctions.

**Discussion**

Autonomic dysfunction is a common feature with high incidence in α-synucleinopathies, including cardiovascular, gastrointestinal,

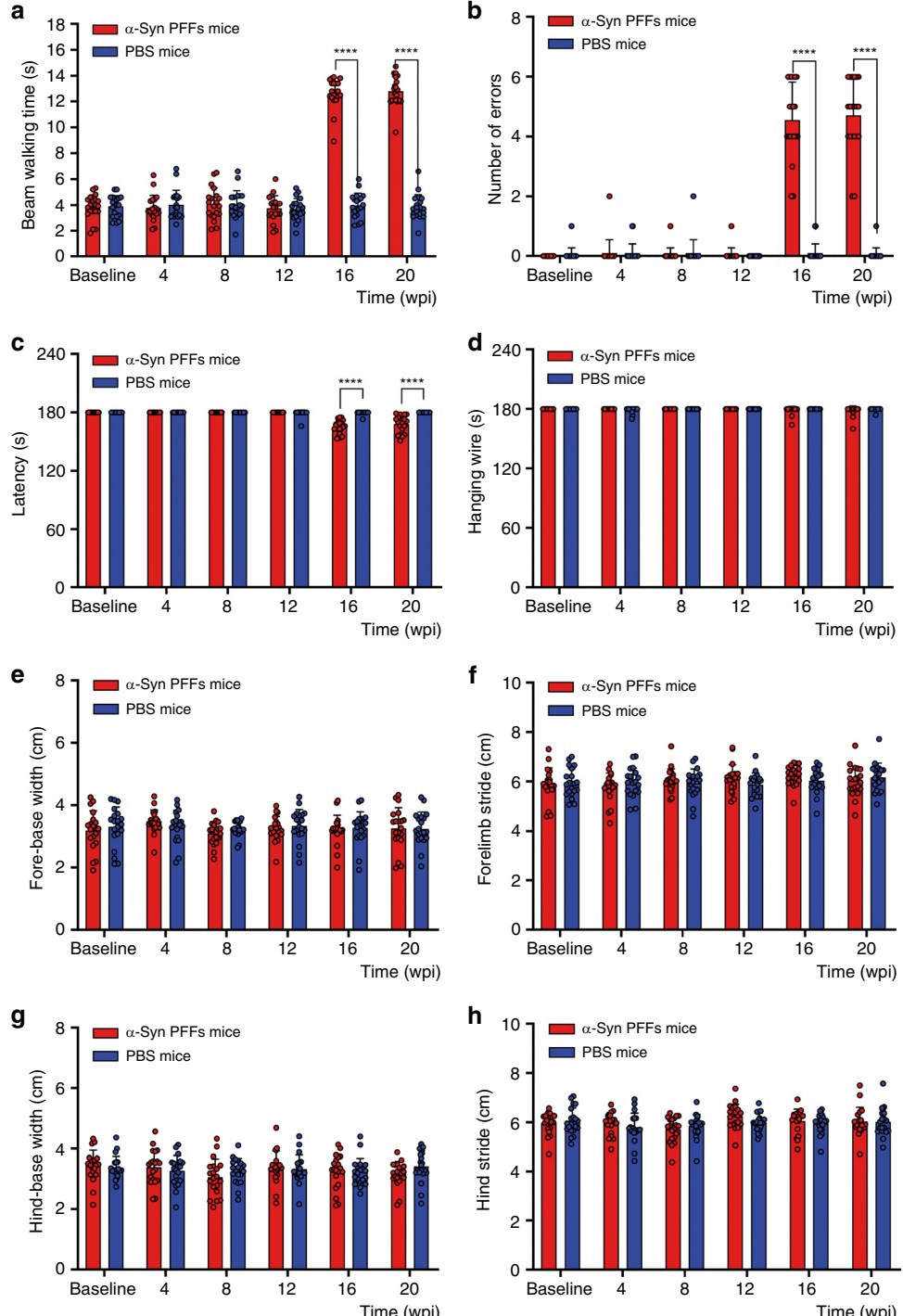

**Fig. 5 Behavioral analysis of α-Syn PFFs mice and PBS mice. a**, **b** The average time to cross the beam (**a**) and the average number of sideslip errors on the beam (**b**). **c** Latency to fall from the rod. **d** The average time in the hanging wire test. **e**, **f** Footprint analysis of the fore-base width (**e**) and the forelimb stride length (**f**). **g**, **h** Footprint analysis of the hind-base width (**g**) and the hindlimb stride length (**h**). n = 20 animals per group. The error bar in all panels represents the standard deviation (SD). Data are the means ± SD. Statistical analysis was performed using the Student's t test, ****P < 0.0001. Source data is available as a Source Data file.

urogenital dysfunctions, and sweating disorders[4,7]. Among the α-synucleinopathies, PAF is a sporadic idiopathic neurodegenerative disorder characterized by gradually progressive severe autonomic failures such as orthostatic hypotension, hypohidrosis, constipation as well as urinary and sexual dysfunctions without any motor dysfunctions[29]. Accordingly, neuropathology confirmed that selective accumulation of α-Syn-positive inclusions localized in the central autonomic pathways and peripheral autonomic nervous system (PANS) including the nuclei associated with autonomic functions in the CNS and autonomic effector organs in PAF[9–15]. To date, the pathogenesis of PAF remains unclear; even the underlying mechanisms of autonomic dysfunction in α-synucleinopathy are largely unknown. To explore the pathogenesis of PAF may provide a breakthrough to better understand the pathophysiological mechanisms of autonomic failures in α-synucleinopathy.

In vitro studies provided the molecular insight of neuron-to-neuron transmission of pathological α-Syn[16,30–32]. Recent experimental studies further confirmed prion-like spreading of α-Syn pathology by exogenous inoculation of α-Syn PFFs in vivo. Intracerebral or peripheral injections of pathological α-Syn initiates progressive α-Syn pathology in the CNS and motor impairments in TgM83[+/−] mice expressing human A53T mutated α-Syn, which is known as the most popular transgenic mouse model to study α-synucleinopathies[18,19,21,33,34]. Sangjune Kim et al.[35] confirmed that the gut-to-brain propagation of pathologic α-Syn via the vagus nerve caused PD-like neuropathology and motor and non-motor symptoms which can be prevented by truncal vagotomy after α-Syn PFFs injection. However, autonomic failures in these mouse models were rarely evaluated. Our study shows that uni- or bilateral injections of α-Syn PFFs into the stellate ganglia and celiac ganglia of TgM83[+/−] mice induce spreading of α-Syn pathology, specifically through autonomic pathways to both the central autonomic pathways central autonomic pathways and PANS (Fig. 6). In the central autonomic pathways, α-Syn-positive profiles mainly located in the central autonomic nuclei at 2 mpi. These findings indicate that α-Syn pathology transmitted from the sympathetic ganglia to the CNS in a retrograde transsynaptic manner. In the PANS, α-Syn pathology is widely distributed in the autonomic effector organs via anterograde axonal spreading, including the heart, stomach, ileum, colon, and skin. In contrast, age-matched PBS mice did not show any neuropathological lesions upon the injections. The data presented here provide the evidence that α-Syn pathology transmits from the autonomic ganglia to the peripheral effector organs in the manner of anterograde axonal transport. Three months after injection, α-Syn PFFs mice gradually developed orthostatic hypotension, constipation, hypohidrosis, and hyposmia without movement impairments, which mainly attribute to the special distribution pattern of α-Syn pathology in the autonomic nervous system. This appearance of α-Syn pathology parallels the onset of the non-motor symptoms. We analyzed the correlation between semi-quantitative grading of α-Syn pathology and non-motor symptoms using Spearman's rank correlation (Supplementary Fig. 2c–f). Data of α-Syn PFFs mice before injection and 1–5 mpi were collected. Our research is based on previous studies, but differs from them in terms of injection position, onset time of α-Syn pathology, pathological α-Syn distributed regions, non-motor symptoms, and motor impairments.

In the CNS, pathological α-Syn deposited in the IML of the spinal cord, 10N, Amb, Sol, LC, Bar, RN, PAG, and Arc in α-Syn PFFs mice. These distinct neuronal populations were confirmed to take part in regulating cardiovascular function[36–38]. At 3 mpi, α-Syn PFFs mice developed orthostatic hypotension with the normal heart rate variability as expected. In addition, the α-Syn PFFs mice suffered severe constipation at 3 mpi, which probably results from the pathological α-Syn deposition in the 10N, IML, Arc, LC, and Bar, which are associated with gastrointestinal function[37]. Furthermore, α-Syn PFFs mice showed notable olfactory dysfunction. TH immunohistochemistry revealed TH-positive neuron and nerve fiber loss without α-Syn pathology in Ob of α-Syn PFFs mice. Previous studies demonstrated that olfactory dysfunction of PAF may be associated with noradrenergic denervation and decreased levels of norepinephrine[39]. We propose that spreading of α-Syn pathology through the sympathetic pathway potentially caused damage to the sympathetic neuronal functions and axonal transport, and resulted in the decreased neurotransmitter synthesis, which was evident by decreased TH-positive neurons in the Ob. Together, these findings potentially indicate the hyposmia in PAF is closely related to the significant decreases in norepinephrine levels in the Ob. The

number of sweating spots in the paws of α-Syn PFFs mice was significantly reduced compared to the PBS mice, most probably owing to the α-Syn pathology in the IML of the spinal cord. Further, the reticular formation exhibits abundant pathological α-Syn in α-Syn PFFs mice, including the raphe nuclei, PCRt, PnC, PnO, and Gi. Similarly, the 7sp and reticular formation were the first affected regions by pathological α-Syn in the CNS of unilaterally α-Syn PFFs-injected mice. The reticular formation is known to have anatomical and functional projections with the autonomic nervous system[40–42]. Nasse et al. previously reported the local circuit inputs to the reticular formation from sol[41]. Neurons in the Gi project directly to sympathetic preganglionic neurons in the IML in the thoracic spinal cord[40]. Reticulospinal tracts arise from the reticular formation and terminate in Laminae VI-IX of the spinal cord[43,44].

In the PANS, pathological α-Syn deposits were detected in the multiple autonomic effector organs in α-Syn PFFs mice at 2 mpi, which may result from the anterograde axonal transmission of pathological α-Syn to nerve terminals of target organs. The orthostatic hypotension might be associated with pathological α-Syn accumulation in the vascular wall. Cardiac autonomic dysfunction caused by pathological α-Syn accumulation in nerve terminals of myocardium induces discordant responses between heart rate and blood pressure in α-Syn PFFs mice. Besides, the pathological α-Syn accumulation in vessel wall results in the ineffective vascular constriction. Therefore, the impaired heart rate responses and abnormal blood pressure in α-Syn PFFs mice may result from the baroreflex dysfunction caused by the deposits of pathological α-Syn in the cardiovascular autonomic pathway. The hypohidrosis was possibly in connection with pathological α-Syn depositing in the sweat glands. The decrease in gastrointestinal transit was related to pathological α-Syn-positive profiles in the gastrointestinal glands. In addition, unilaterally α-Syn PFFs-injected mice also developed various autonomic dysfunctions including orthostatic hypotension, constipation, hypohidrosis, and hyposmia. Therefore, based on the evidence from the current study, it is most likely that the pathological α-Syn deposition in the autonomic effector organs exacerbated the autonomic symptoms in α-Syn PFFs mice.

Abnormal gastrointestinal motility in the α-Syn PFFs mice might not only be related to impaired sympathetic pathways but also to impaired parasympathetic pathways. To evaluate whether α-Syn pathology located in 10N is caused by transmission of pathologic α-Syn via the vagus nerve in α-Syn PFFs mice, we further cut off the vagus nerve distributed near the gastrointestinal tract and injected α-Syn PFFs to celiac ganglia 3 days later. We found that there was no α-Syn pathology in 10N of α-Syn PFFs mice after truncal vagotomy even at 7 mpi (Supplementary Fig. 3). These mice still showed significant α-Syn pathology in other regions and nuclei as early as 2 mpi. We infer that the pathological α-Syn from the sympathetic ganglia could spread to the nerve terminals in the heart, blood vessels, and gastrointestinal tract, where it is taken in by the vagus nerve and further spreads to 10N transsynaptically retrogradely.

Besides PBS-injected TgM83[+/−] mice, we also used α-Syn monomer-injected TgM83[+/−] mice as another control group. In addition, we also performed this experiment in C57BL/6 wild-type background mice. No pathological change or phenotype was detected in the α-Syn monomer-injected TgM83[+/−] mice even at 7 mpi (Supplementary Fig. 2o–t). In α-Syn PFFs or PBS-injected C57BL/6 wild-type mice, there is no pathological α-Syn or phenotype induced (Supplementary Fig. 2u–z). Previous studies provide the evidence that there might be a requirement of homogenous α-Syn transgene to induce the formation and transmission of α-Syn aggregates[45]. Exogenous injection of human α-Syn PFFs may not be able to initiate the α-Syn

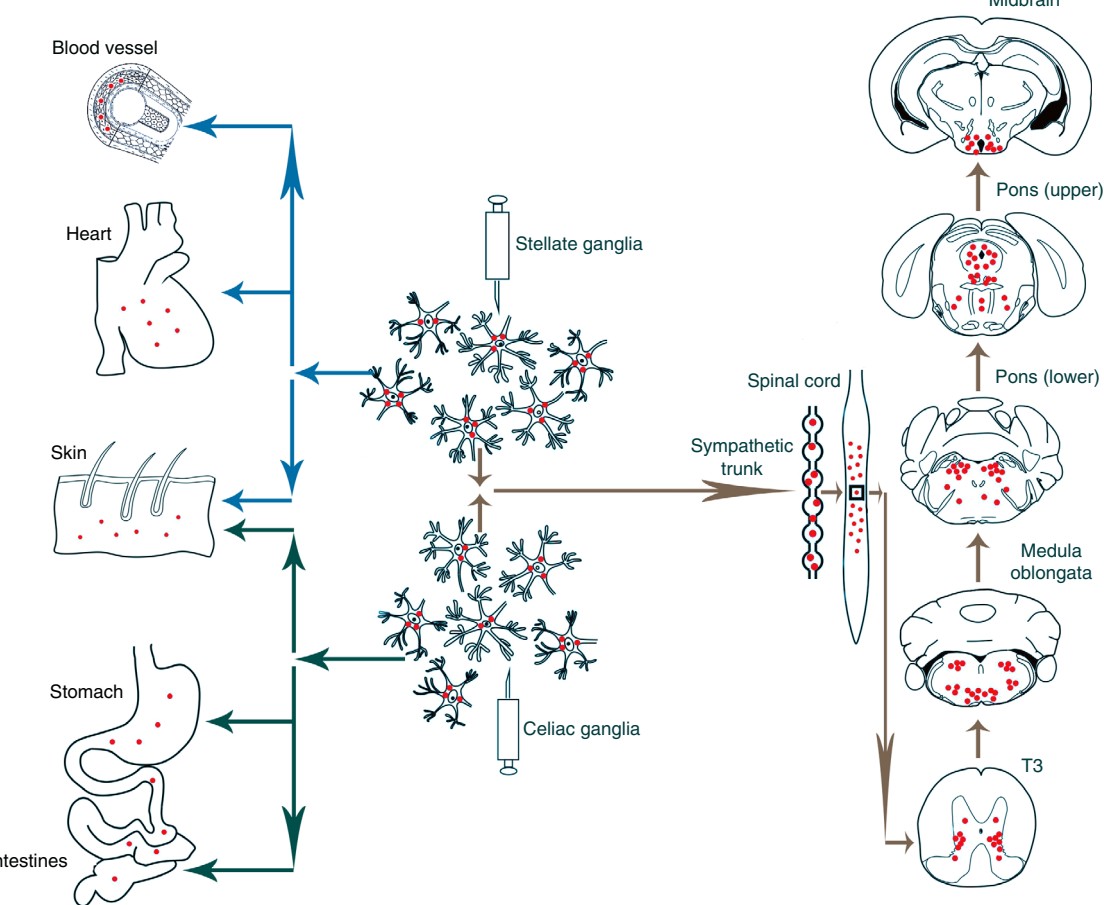

**Fig. 6 The Schematic displaying the spreading pattern of pα-Syn in autonomic nervous system.** α-Syn pathology transmits in multiple regions in central and peripheral nervous system including the gray matter of the spinal cord, medulla oblongata, pons, midbrain, myocardium, blood vessel, gastrointestinal tract, and skin, following inoculation in stellate ganglia and celiac ganglia. α-Syn: α-synuclein; PAF: pure autonomic failure; PFFs: preformed fibrils; CNS: central nervous system; PD: Parkinson's disease; DLB: dementia with Lewy bodies; MSA: multiple system atrophy; RN: raphe nucleus; IHC: immunohistochemistry; IF: immunofluorescence; mpi: months post-injection; Arc: arcuate hypothalamic nucleus; PAG: periaqueductal gray; Sol: nucleus of the solitary tract; Ob: olfactory bulb; TH: Tyrosine hydroxylase; TPH: tryptophan hydroxylase; ChAT: choline acetyltransferase; PBS: phosphate-buffered saline; MAP-2: Microtubule-associated protein-2; GFAP: glial fibrillary acidic protein; MBP: myelin basic protein; Ub: ubiquitin; SBP: systolic blood pressure; DBP: diastolic blood pressure; wpi: weeks post-injection; PANS: peripheral autonomic nervous system; PVDF: polyvinylidene fluoride; SGs: sweat glands; SD: standard deviation; 3Sp: lamina 3 of the spinal gray; 4Sp: lamina 4 of the spinal gray; 5SpL: lamina V of the spinal gray, lateral part; 5SpM: lamina V of the spinal gray, medial part; 7Sp: lamina VII of the spinal gray; 8Sp: lamina VIII of the spinal gray; 10N: dorsal motor nucleus of vagus 10Sp: lamina X of the spinal gray; Amb: ambiguus nucleus; APTD: anterior pretectal nucleus, dorsal part; ArcLP: arcuate hypothalamic nucleus, lateroposterior part; ArcMP: arcuate hypothalamic nucleus, medial posterior part; ATg: anterior tegmental nucleus; Ax9: axial muscle motoneurons of lamina 9; Bar: Barrington's nucleus; CGA: central gray, alpha part; CI: caudal interstitial nucleus of the medial; CGPn: central gray of the pons; DLG: dorsal lateral geniculate nucleus; DLPAG: dorsolateral periaqueductal gray; DMPAG: dorsomedial periaqueductal gray; DMPn: dorsomedial pontine nucleus; DMSp5: dorsomedial spinal trigeminal nucleus; DMTg: dorsomedial tegmental area; DRD: dorsal raphe nucleus, dorsal part; DRV: dorsal raphe nucleus, ventral part; DRVL: dorsal raphe nucleus, ventrolateral part; DTgC: dorsal tegmental nucleus, central part; DTgP: dorsal tegmental nucleus, pericentral part; Gi: gigantocellular reticular nucleus; GiA: gigantocellular reticular nucleus, alpha part; ICL: intergeniculate leaf; ICo9: intercostal muscle motoneurons of lamina 9; IML: intermediolateral column; IRt: intermediate reticular nucleus; LC: locus coeruleus; LDCom: lumbar dorsal commissural nucleus; LDTg: laterodorsal tegmental nucleus; lfp: longitudinal fasciculus of the pons; LPAG: lateral periaqueductal gray; LPGi: lateral paragigantocellular nucleus; LPMC: lateral posterior thalamic nucleus, mediocaudal part; LPMR: lateral posterior thalamic nucleus, mediorostral part; LSp: lateral spinal nucleus; ml: medial lemniscus; mlf: medial longitudinal fasciculus; MnR: median raphe nucleus; O: nucleus O; PCRt: parvicellular reticular nucleus; PCRtA: parvicellular reticular nucleus, alpha part; PF: parafascicular thalamic nucleus; PMD: premammillary nucleus, dorsal part; PMnR: paramedian raphe nucleus; PMV: premammillary nucleus, ventral part; Pn: pontine nuclei; PnC: pontine reticular nucleus, caudal part; PnO: pontine reticular nucleus, oral part; Po: posterior thalamic nuclear group; PR: prerubral field; Pr5DM: principal sensory trigeminal nucleus, dorsomedial part; Pr5VL: principal sensory trigeminal nucleus, ventrolateral part; PSTh: parasubthalamic nucleus; py: pyramidal tract; Rbd: rhabdoid nucleus; RC: raphe cap; RMg: raphe magnus nucleus; RtTgP: reticulotegmental nucleus of the pons, pericentral part; RVL: rostroventrolateral reticular nucleus; SGl: superficial glial zone of the cochlear nuclei; SolDM: nucleus of the solitary tract, dorsomedial part; SolIM: nucleus of the solitary tract, intermediate part; SolM: nucleus of the solitary tract, medial part; SolV: solitary nucleus, ventral part; SolVL: nucleus of the solitary tract,ventrolateral part; Sp5I: spinal trigeminal nucleus, interpolar part; Sph: sphenoid nucleus; ThAb9: thoracoabdominal wall muscle motoneurons of lamina 9; ts: tectospinal tract; Tz: nucleus of the trapezoid body; VCA: ventral cochlear nucleus, anterior part; VLPAG: ventrolateral periaqueductal gray; xscp: decussation of the superior cerebellar peduncle.

propagation and clinical phenotypes in C57BL/6 mice owing to the lack of human α-Syn in mice.

PAF is a very rare and slowly progressive α-synucleinopathy characterized by PAF such as orthostatic hypotension, hypohidrosis, constipation as well as urinary and sexual dysfunctions without motor dysfunctions[29]. To date, the pathogenesis of PAF remains unclear and there is currently no specific treatment to stop its progression[8], nor are there available animal models to study PAF. PAF is featured by selective accumulation of α-Syn-positive inclusions limited within the autonomic nervous system. Postmortem studies revealed that Lewy bodies were detected in the sympathetic ganglia, IML of the spinal cord, 10N, LC, substantia nigra, raphe nucleus, and brainstem reticular formation in patients with PAF[9–15]. The pathological α-Syn distribution of the α-Syn PFFs mice in the present study fits with postmortem studies of PAF patients to a large extent (Supplementary Table 2). Here, we established a mouse model exhibiting PAF-like neuropathology and progression of autonomic symptoms. Moreover, our findings provide the evidence that the formation and propagation of α-Syn-positive inclusions along the autonomic pathways result in PAF-like neuropathology and clinical syndromes. However, there are still some defects in our research. Firstly, the α-Syn PFFs mice died within a short time once the phenotypes appeared, so we fail to observe the progressive autonomic dysfunctions in diseased mice. Secondly, further studies are needed to explore the underlying mechanisms and potential therapeutic targets of autonomic dysfunction in α-synucleinopathy in wild-type mice by injecting mouse α-Syn PFFs instead of human α-Syn PFFs.

In conclusion, our data here demonstrate that the local injection of α-Syn PFFs in sympathetic ganglia specifically induces progressive α-Syn pathology in the autonomic nervous system and the corresponding autonomic dysfunctions without motor dysfunction in vivo. These findings also have key implications for understanding the pathogenesis of autonomic dysfunction caused by α-synucleinopathy. This de novo mouse model should be useful and valuable for exploring disease-modifying therapies for pathological α-Syn induced autonomic dysfunctions.

## Methods

**Animals**. Hemizygous M83 transgenic mice overexpressing human α-Syn with the A53T mutation (TgM83[+/−] mice) driven by the mouse prion protein promoter as well as endogenous mouse α-Syn[33] were purchased from Nanjing Biomedical Research Institute of Nanjing University (Nanjing, China). Mice were provided with free access of food and water under the condition of a 12 h light/dark cycle. Each mouse was given 3 to 7 g/day of feed, and 4 to 7 ml/day of purified water. All mouse procedures were conducted in accordance with the Guide for the Care and Use of Laboratory Animals. The protocols and procedures for the study were approved by Institutional Ethics Committees of Zhengzhou University.

**Preparation and characterization of α-Syn PFFs**. α-Syn (S-1001, rPeptide) was resuspended in assembly buffer (20 mM Tris-HCl, pH 7.4, 100 mM NaCl) at a concentration of 1 mg/ml. The mixture was placed in 2 ml sterile polypropylene tubes, sealed with parafilm. It was then agitated by a magnetic stirrer apparatus (MS-H-Pro+, Scilogex, China) at 350 rpm and 37 °C for seven days. Then, the α-Syn fibrils were sonicated by an ultrasonic cell disruptor at 10% of its peak amplitude (Scientz-IID, Ningbo, China) for 45 s. Sequentially, the prepared α-Syn PFFs were immediately frozen and stored in a −80 °C freezer. The nature of the α-Syn PFFs forms was examined using a JEOL 1400 transmission electron microscope. It was adsorbed onto carbon-coated 200-mesh grids and negatively stained with 1% uranyl acetate. The grids were required to be dry among all steps and examined using a Gatan Orius CCD camera (Gatan). The fibril nature of the α-Syn PFFs we generated was confirmed by analysis of proteinase K (PK) resistance using a dot-blot test. PK (2.0 μg/ml) (Solarbio, China) was used (Supplementary Fig. 1e).

**Modeling surgery**. Male TgM83[+/−] mice (2 months of age) were anesthetized with isoflurane inhalation and fixed in a supine position. After skin preparation, an anterior cervical incision was made in the middle to expose the clavicle. The stellate ganglia are located on the dorsal side of clavicle and sternum, anterior to the 7th cervical spinal cord transverse process and attached to brachial plexus (Supplementary Fig. 1a). α-Syn PFFs (1.0 mg/ml) or PBS were injected bilaterally into the stellate ganglia of mice in the model group and control group, respectively, (1.5 μl per side for a total of 3.0 μl) under the operating microscope. The injection was performed using a 10-μl Hamilton syringe with a 34-gauge needle at a rate of 0.5 μl/min. The syringe remained at the injection site for a further 5 min and then the incision was sutured after it was sealed with glycerol. A ventral midline incision was made in the abdomen to expose the celiac artery and superior mesenteric artery. The celiac ganglia are located on the ventral side of the descending aorta, the medial side of the kidney, the head of the superior mesenteric ganglion, and embedded in adipose tissue (Supplementary Fig. 1b). The celiac ganglia injection was performed as described previously. Under the operating microscope, α-Syn PFFs (1.0 mg/ml) or PBS were injected into bilateral celiac ganglia of mice (4.0 μl per side for a total of 8.0 μl). To confirm a successful injection into the stellate ganglia and celiac ganglia and the volume that could be injected without substantial damage or leakage, trypan blue was injected into the ganglia prior to injection of α-Syn PFFs or PBS (Supplementary Fig. 1a, b). It was determined that 1.5 μl for the stellate ganglion and 4.0 μl for the celiac ganglion could be safely injected under the operating microscope. We also performed unilateral intra-ganglion injections in the right stellate and celiac ganglion in some other TgM83[+/−] mice. Mice, after injection, were housed under standard conditions. The pathology and functions of the mice were detected after the injection in a series of time points.

**Immunohistochemical and immunofluorescent analyses**. Brain, spinal cord, bilateral stellate ganglia and celiac ganglia, blood vessel, heart, stomach, and intestines from perfused mice were fixed in a mixture of 30% sucrose solution and 4% paraformaldehyde. The paraffin-embedded tissues were cut into 4 μm thick sections with a Rotary Microtome (Leica RM2235, Leica, Nussloch, Germany). Then, sections were mounted on glass slides, dehydrated and deparaffinized[46,47]. Nonspecific sites of antigens were blocked with normal sera of the host animal species of the secondary antibodies. Slides were incubated with the primary antibody overnight at 4 °C. We detected the bound antibodies by a Streptavidin-Peroxidase kit (Bioss, China) and visualized them using 3-3′-diaminobenzidine (DAB; Neobioscience). The slides were counterstained with hematoxylin. For the immunofluorescence analysis, the sections were incubated with the primary antibodies overnight at 4 °C[48]. After rinsing three times with PBS, slides were incubated with secondary antibodies. Then, cell nuclei were stained using Hoechst33258 (1:1000, Solarbio). Finally, the slides were coverslipped with fluoromount and observed using a fluorescence microscope. Serial coronal sections of the medulla, pons, midbrain, and olfactory bulb were analyzed, especially the four representative sections of medulla (Bregma −6.72 mm), pons (lower, Bregma −5.40 mm), pons (upper, Bregma −4.48 mm), and midbrain (Bregma −2.48 mm). The distance between medulla (Bregma −6.72 mm) and pons (lower, Bregma −5.40 mm) is about 1.32 mm, between pons (lower, Bregma −5.40 mm) and pons (upper, Bregma −4.48 mm) is about 0.92 mm, and between pons (upper, Bregma −4.48 mm) and midbrain (Bregma −2.48 mm) is about 2.00 mm. Commercially available antibodies used are shown in the Supplementary Table 1. The immunohistochemical images were processed using ImageJ software (US National Institutes of Health) and the percentage of areas occupied by pα-Syn and TH immune-positive staining were quantified.

**Tissue preparation**. Four-month-old animals were anesthetized and decapitated, and tissues of mice were prepared. Tissues were separated into detergent-soluble and -insoluble fractions by TBS + (50 mM Tris-HCl, pH 7.4, 175 mM NaCl, 5 mM EDTA), and complete protease inhibitor mixture (Thermo Fisher Scientific, U.S.A). After 30 min of ultracentrifugation at 120,000g at 4 °C, the resulting supernatants representing the soluble fractions were collected, and then, the pellets were further rinsed in TBS + containing 1% Triton X-100, TBS + containing 1 M sucrose, and RIPA buffer (Beyotime, China) by 20 min of ultracentrifugation at 120,000g at 4 °C separately. The final pellets representing the insoluble fractions were solubilized in 8 M urea/5% SDS.

**Western blot**. Ten microgram of each fraction was loaded and electrophoresed on 12% or 15% SDS-polyacrylamide gel and transferred to polyvinylidene fluoride (PVDF) membranes (Millipore, Germany). After blocking with 10% nonfat dry milk in TBS-T for 1 h at room temperature, the membranes were incubated with primary antibodies- phospho-α-Syn (Ser 129) (rabbit, Abcam, 1:2000) over two nights. The membranes were then washed in TBST (TBS with 0.1% Tween-20) several times, then incubated with HRP-conjugated secondary antibody for 2.5 h at room temperature and visualized with enhanced chemiluminescence (Thermo Fisher Scientific, U.S.A). Proteins on the blots were normalized to GAPDH (mouse, Millipore,1:2000). The most important full-size western blots are displayed in Supplementary Fig. 4.

**Evaluation of autonomic function**. The mice were evaluated by the following tests to assess the autonomic function once a month before and after injection.

Systolic and diastolic blood pressure of mice were measured for fifteen cycles by a noninvasive tail-cuff BP analyzer (CODA system, Kent Scientific, Torrington, CT)[49–51]. We examined that the tubes had no air leaks before the measurements began[52]. Mice were placed in the restrainer for 10–15 min each time under quiet and warm conditions, commonly controlling the temperature of mouse tail to

around 32–35 °C prior to the measurement. The mice were placed on heating pads to ensure sufficient blood flow to the mouse tail. It was of importance to schedule them for approximately the same time each day. Additionally, the mice were put in the upright position and lying position separately to detect whether orthostatic hypotension was apparent. The mice were in the upright position for 10 min and then the orthostatic blood pressure was measured. The data was collected and averaged by the software connected to the computer.

The buried food test was an effective method to evaluate the olfactory function[53–55]. Mice were given cookies and food-restricted on a diet for 16 h prior to the experiment. Generally, we could put the cookies on the surface and bury the cookies under the bedding. The cookies were buried approximately 3 cm under the new bedding in the test cage at random locations. The mice were placed in the center of cage after all preparations. Then, we recorded the latency to dig up the cookies. If the latency was beyond 5 min, we recorded a maximum time of 5 min[56]. If the latency was much longer, olfactory function would become weak.

Mice were fasted for 16 h but not deprived of water before the test. 10% activated carbon was administered by gavage to the prepared mice. The mice were placed in clean cages to observe the appearance of black feces. The interval from the oral administration to the appearance of black feces reflected the gastrointestinal function. The mice received oral administration of 10% activated carbon and after 30 min, the mice were anesthetized with isoflurane and then sacrificed by cervical dislocation. The intestine from the pylorus to the blind intestine in both groups of mice were dissected and removed[57]. The gastrointestinal transit (GI%) was calculated as the percentage of the distance traveled by the activated carbon relative to the total length of the small intestine. GI transit (%) = (distance traveled by the activated carbon)/ (total length of the small intestine) × 100%[57]. The gastrointestinal transit and the defecation time were the main standard to evaluate the gastrointestinal function.

Mice were anesthetized with isoflurane inhalation and placed in the prone position. Then, the hindpaws were cleaned and dried, smeared in 3.5% iodine solution. After it dried, starch was painted on the bilateral forepaws and hindpaws. Subcutaneous injection of pilocarpine to the back of the mice (0.035 ml/10 g) stimulated sweating of the mice. The images of the spots of the sweat glands (SGs) reactive to pilocarpine stimulation on the plantar surface of mice hindpaws were taken by digital camera approximately 5–15 min after the injection. Finally, the number of the sweat spots in the mouse bilateral forepaws and hindpaws was counted to assess the sweating function[58,59].

**Behavioral evaluation**. The mice were evaluated by the following tests to assess the motor function once a month before and after injection.

We evaluated the activity of the mice with a rotating rod (Rotarod YLS-4C; YiYan Science and Technology Development Co., Ltd. Shandong, China). Mice were placed on the rod rotating each time at 30 rpm. We recorded the latency to remain on the rotarod. Latency greater than 180 s (s) was recorded as 180 s. With a 15-min inter-trial interval, the mice were tested three times per day. Finally, the latency was averaged.

The gait of mice was examined by the footprint test. All of the mice were painted on the paws with water-soluble non-toxic dyes (forepaws in red and hindpaws in green). The mice walked along a restricted cardboard tunnel (50 cm long, 5 cm wide, 10 cm high) lined with white paper (42 cm long, 4.5 cm wide) into an enclosed box and then a set of footprints were finished. Three footsteps from the middle portion of each walk run were measured for the following parameters (cm): (1) stride length (front and hind legs). (2) The front-base and hind-base width. The average of each set of values was analyzed[60].

The hanging wire test was performed to evaluate the general muscular strength. Mice were placed on a horizontally positioned screen with grids. The latency to leave the screen was recorded. The time the mouse left the screen was up to 180 s. The average latency of three trials per day was analyzed.

The Beam walking test was employed to examine the balance of the mice. The beams were 50 cm parallel to the floor including two types of wood (80 cm long, one was 1.6 cm, and the other 0.9 cm wide). Two tests for training were performed with the 1.6 cm width beam. Then, mice were tested with the 0.9 cm width beam. The latency to transverse 50 cm and the number of sideslip errors of three trials were averaged and analyzed.

**Statistical analysis**. All of the data are expressed as mean ± standard deviation (SD). Mice were divided into the α-Syn PFFs group and the PBS group. Paired or unpaired Student's $t$ test was performed to compare the IHC results and the heterogeneity of function evaluation between the α-Syn group and PBS group. The survival curve was calculated using the log-rank test and Gehan–Breslow–Wilcoxon test. $P \leq 0.05$ indicates that the difference between the groups is statistically significant. Data from IHC and negative-stained transmission electron micrographs were analyzed by ImageJ software (US National Institutes of Health). The correlation between semi-quantitative grading of α-Syn pathology and non-motor symptoms was analyzed using Spearman's rank correlation. Statistical analysis of sonicated α-Syn PFFs was performed using SPSS 21.0 (IBM, Armonk, New York, USA). Data of Western blot, function evaluation, Spearman's rank correlation, and the survival curve were analyzed by the Prism software 8.0 (GraphPad Software, La Jolla, CA).

**Reporting summary**. Further information on research design is available in the Nature Research Reporting Summary linked to this article.

## Data availability
All relevant data supporting the findings of this study are either included within the article and its Supplementary Information files or are available upon request from the corresponding author. Source data underlying Figs. 2, 4, 5 and Supplementary Figs. 1 and 2 are available as a Source Data file.

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

## Acknowledgements

This work was supported by grants from the National Natural Science Foundation of China (No. 81671267, 81471307, 81301086, 81430023, 81430025 and U1801681) and the Key Field Research Development Program of Guangdong Province (2018B030337001). Acknowledgements were also to the supports of the Swedish Research Council (K2015-61X-22297-03-4; 2019-01551), EU-JPND (aSynProtec) and EU-JPND (REfrAME), EU H2020-MSCA-ITN-2016 (Syndegen). We thank all our collaborators at the First Affiliated Hospital of Zhengzhou University for their assistance with patient sample collection. Specially, we thank Prof. Guanghui Wang for his excellent technical support. We thank Dr. Andrew McCourt for linguistic editing.

## Author contributions

X.W. conceived and designed the experiments; X.W., X.D. and M.M. coordinated the whole project; L.Z., X.J. and M.H. conducted the histological preparations; X.J., M.H. and R.K.F. conducted modeling surgery and behavioral test; L.Z., M.H., J.T. and J.Y. performed immunostainings and western blot experiments; E.W., J.Y.L. and X.D. provided statistical analysis and technical support; X.W., X.D., M.M., L.Z., X.J., M.H., B.T. and J.Y.L. participated in final data analysis and interpretation; X.W., E.W., L.Z., M.H. and X.J. did most of the writing with input from other authors. All of the authors discussed the results and commented on the manuscript.

## Competing interests

The authors declare no competing interests.
