## [Peer Review File · Nature Communications]

Reviewers' comments:

Reviewer #1 (Remarks to the Author):

This is a very interesting work by Wang and colleagues that try to generate a mouse model with autonomic dysfunction. Although the idea for the study, in principle, is interesting and highly relevant, there are some concerns about the study and conclusions that are listed below.

Major:

- The authors refer to central or peripheral Autonomic nervous system in the text. Autonomic nervous system is an efferent system and because of that, it is only part of the peripheral nervous system. In my opinion, it is confusing to classify neurons that are presympathetic or pre-ganglionic as central autonomic nervous system.
- During the cardiovascular test, the results do not show changes in heart rate. How do the authors explain that, if the heart rate would be the responsible for cardiac output and consequently by the changes in blood pressure? Is possible that the reduction in the blood pressure be a result of a vascular instead of baroreflex dysfunction?
- The immunohistochemical experiments should be performed with double staining to conclude which neuronal population was affected. Some examples: TH + Syn for staining C1 and A6 regions; Choline acetyltransferase + Syn for staining nucleus Ambiguus, dorsal motor nucleus of vagus and preganglionic neurons; tryptofan-hydroxylase + Syn for staining raphe.
- The authors performed the injections into sympathetic ganglia. So, why is possible to see the presence of Syn in parasympathetic preganglionic neurons as NA and DMV? How can the authors justify the changes in gastrointestinal motility if only the sympathetic pathway is affected?
- It is surprising do not see the presence of Syn in C1 RVLM, especially C1 region, since those neurons are presympathetic and involved in the control of vascular resistance and heart rate.
- The authors need to provide more information in the Methods section about antibodies, the number of sections analyzed per brain, the thickness of the sections and the distance between the sections analyzed.

Minor:

- I would expect some kind of progressive functional responses. It looks that as soon as the disautonomy appears, it is total
- The authors perform injections into the ganglion. However, it is not clear how the authors know that they reach the ganglion.
- It would be interesting to describe the advantages and disadvantages of this model
- Figure 2: which region of the medulla was analyzed?
- Figure 4K: which is the staining for TH? Where is the Syn?
- Why do the animals that receive Syn injection die earlier?
- How was the drink and food intake?

Reviewer #2 (Remarks to the Author):

Xue-Jing Wang et al. present an interesting story about autonomic neuropathology and phenotype induced by autonomic ganglionic injection of α -syn PFFs. The idea is novel, and the quality of majority data looks nice. Some experiment design and rationale are quite weak, which significantly hamper the enthusiasm on this manuscript. Some experimental details are missing, but are really big concerns for the significance of the manuscript. I would suggest the authors need to revise the manuscript carefully with great effort.

The main weakness of the manuscript is to use the α -syn transgenic mice, which is limited to familial Parkinson's disease. The TgM83 mice start to die from 7-month old, however the experiment design is to sacrifice the mice 7-months after the injection, which has been 9 months old (2-month old receive PFFs injection). The variation of TgM83 in the survival curve may have

interfered the conclusion. TgM83 lines express both human A53T and endogenous mouse α -synuclein, and A53T mutant α -syn and mouse α -syn definitely are distinct strain from inoculated human wildtype α -syn PFFs (S-1001, rpeptide). Furthermore, it is not clear that the autonomic dysfunction is caused by the inoculation of PFFs into stellate and celiac ganglia, since aged α -syn transgenic mice may have some non-motor symptoms and even pathology in the gut as well (PMID: 20106867). What about aged TgM83, do they have the some similar autonomic dysfunction or phenotype?

Supplementary Fig. 1B showed the size of α -syn PFFs is very long (> 1000 nm), which is quite different from the size (< 100 nm) in the published papers, please see PMID: 23161999, 27708076. There are some other lowercase letters, which are confusing.

Supplementary Fig. 2A, the mouse number is missing, but based on the dots in the plot, there are only 5 mice for each group, although the two groups may exhibit dramatic difference in the survival curve. Fig. 2B, how many mice are included in the result, and what is the error bar? How well these non-motor syndrome can be characterized in mouse model? Is there any well-established paper for citation? Is there any correlation between the non-motor symptoms appeared in different timeline and the alpha-synuclein pathology appeared in different timeline? Does the observed pathology pattern fit in the PAF?

Fig. 2C, the quality of the pathology is too low, high magnified images should be provided.

Some references should be cited, including LAG3 as the receptor of α -syn PFFs (PMID: 27708076), gut-PFFs model (Neuron 2019 in press).

The hypohidrosis results are related to number of paws or number of animal?

Since the experiment design contains several uncertain factors, which significantly reduce the quality of the discovery, and hamper us to effectively studying the role of α -syn PFFs in α -syn spreading from the stellate and celiac ganglia.

All the histogram should be shown with scattered dots, to understand the distribution of the raw data. Power analysis is missing. All the experiments should be repeated with three independent times. Mice number in each experiment may need >3 mice at least; 3 mice may be good if power analysis results show the reasonability.

Fig. 2, why there are some overlap with GFAP and pa-syn, is it the phenotype of PAF?

Fig. 3, quantification is missing. Big scale of images should be shown to understand the distribution.

What is site of the phosphorylated α -syn or pa-syn, which is not described clearly in the manuscript?

All the symptoms including hyposmia, hypohidrosis, constipation, and orthostatic hypotension is the non-motor symptoms of Parkinson's disease (PD), however, the authors failed to show the motor-related symptoms as well as the loss of dopaminergic, which may be caused by short incubation time due to TgM83 mice (see WT mice in Neuron 2019 in press)? This is a weakness, or may suggest that it is not a good disease model? What about dementia?

What about the endotoxin, which has been suggested playing an important role in the α -syn strain formation and the pathogenesis of α -syn spreading? The symptoms observed in the manuscript could be attributed by the endotoxin. The good control is α -syn monomer.

Reviewer #3 (Remarks to the Author):

The main objective of the study by Wang et al was to develop a model of pure autonomic failure (PAF) without motor dysfunction as has been the case with other models of alpha-synucleinopathies that target the striato-nigral system. For this purpose, Wang et al inoculated of α -Syn preformed fibrils (PFFs) into stellate of M83 mice a transgenic model expressing low levels of α -Syn. They show that the α -Syn pathology disseminated through the autonomic pathway to the CNS without motor features. The mice also develop functional deficits such as hypotension, constipation and hyposmia. They conclude that their mouse reflects PAF and will help better understand alpha-synucleinopathies with dysautonomia.

This is an interesting study in that this is a relatively new model system of alpha-synucleinopathy with dysautonomia, however several issues need clarification:

1. A number of similar PFF models have been previously developed injecting α -Syn in the periphery developing autonomic deficits. The authors need to clarify in more detail what is different about their model compared to others. It is not completely clear why this model worked differently than other similar models.
2. Need to provide more detailed description and characterization of the PFF used, could it be possible that the animals developed the selective pathology because of a unique strain of α -Syn PFF was used? For example expand on their EM analysis of the PFF showing if they have a ribbon, helical or other structures similar to what has been shown in models mimicking MSA vs PD. Perform analysis of PK resistance vs degradation or PCMA like analysis.
3. The double labeling image in Figure 2 is very low power and difficult to appreciate the cellular localization of the α -Syn. The patches stained with MBP are unclear what they represent, fiber bundles?, they also need to show double label ubiquitin with cellular marker to understand better the cellular localization, the α -Syn aggregates in red only occasionally appear to co-localize with MAP2 and ubiquitin but nothing else, what is the cellular localization of other aggregates? Are they outside the cells?
4. Did they try this experiment in the WT background? Why there is a need of low levels α -Syn tg to express, this means that their model requires templated seeding with homologous α -Syn (human) and not with mouse α -Syn as it has been shown in the original studies of Luk et al. Is there expression of human α -Syn in the tg model in the periphery, they need to characterize. Is the effects dependent on the expression of the transgene in the autonomic nuclei.
5. It will be important to show that the spreading of the pathology is blocked by blocking the vagal nerve, otherwise the spreading mechanisms is unclear given that seeding is occurring in the background or a α -Syn tg model.
6. It would be important to compare to all the relevant controls to better understand to what extent the pathology and functional deficits are the results of synergy between the PFF and the M83 transgene or additive effects or unique to the PFF. The authors only show data on M83 PBS vs PFF, but would be important to compare to WT PBS vs PFF.
7. Need more detailed statistical analysis and to provide in the figure legends the n number per group. Only males were used, what is the explanation, why they didn't consider sex differences?

REVIEWER #1

This is a very interesting work by Wang and colleagues that try to generate a mouse model with autonomic dysfunction. Although the idea for the study, in principle, is interesting and highly relevant, there are some concerns about the study and conclusions that are listed below.

Major:

- The authors refer to central or peripheral Autonomic nervous system in the text. Autonomic nervous system is an efferent system and because of that, it is only part of the peripheral nervous system. In my opinion, it is confusing to classify neurons that are presympathetic or pre-ganglionic as central autonomic nervous system.
- During the cardiovascular test, the results do not show changes in heart rate. How do the authors explain that, if the heart rate would be the responsible for cardiac output and consequently by the changes in blood pressure? Is possible that the reduction in the blood pressure be a result of a vascular instead of baroreflex dysfunction?
- The immunohistochemical experiments should be performed with double staining to conclude which neuronal population was affected. Some examples: TH + Syn for staining C1 and A6 regions; Choline acetyltransferase + Syn for staining nucleus Ambiguus, dorsal motor nucleus of vagus and preganglionic neurons; tryptofan-hydroxylase + Syn for staining raphe.
- The authors performed the injections into sympathetic ganglia. So, why is possible to see the presence of Syn in parasympathetic preganglionic neurons as NA and DMV? How can the authors justify the changes in gastrointestinal motility if only the sympathetic pathway is affected?
- It is surprising do not see the presence of Syn in C1 RVLM, especially C1 region, since those neurons are presympathetic and involved in the control of vascular resistance and heart rate.
- The authors need to provide more information in the Methods section about antibodies, the number of sections analyzed per brain, the thickness of the sections and the distance between the sections analyzed.

Minor:

- I would expect some kind of progressive functional responses. It looks that as soon as the disautonomy appears, it is total.
- The authors perform injections into the ganglion. However, it is not clear how the authors know that they reach the ganglion.
- It would be interesting to describe the advantages and disadvantages of this model.
- Figure 2: which region of the medulla was analyzed?
- Figure 4K: which is the staining for TH? Where is the Syn?
- Why do the animals that receive Syn injection die earlier?
- How was the drink and food intake?

Response to REVIEWER #1

Response (R): Thank you very much for your time and the positive comments. We greatly appreciate your constructive suggestions for improving our manuscript. We have carefully addressed your concerns and advice.

Major:

1. The authors refer to central or peripheral Autonomic nervous system in the text. Autonomic nervous system is an efferent system and because of that, it is only part of the peripheral nervous system. In my opinion, it is confusing to classify neurons that are presympathetic or pre-ganglionic as central autonomic nervous system.

R: We appreciate your comment. Generally, the autonomic nervous system is regarded as a part of the peripheral autonomic system. Postganglionic neurons locate in the autonomic ganglia as a part of the peripheral autonomic nervous system. The cell bodies of the preganglionic sympathetic neurons are exclusively found in the intermediolateral nucleus (IML) of the C8/T1 to L2, while the cell bodies of the preganglionic neurons of parasympathetic nervous system locate in the dorsal motor nucleus of the vagus in the medulla oblongata and sacral parasympathetic nuclei in sacral spinal cord (PMID: 28012041, 7056993) [1, 2]. Besides, the reflex responses of the sympathetic and parasympathetic preganglionic neurons are directly driven by premotor neurons of the CNS (PMID: 29869180) [3]. Therefore, the preganglionic neurons of both sympathetic and parasympathetic pathways and the regulating nuclei locate in the central nervous system. Therefore, we define the central part in the autonomic nervous system as the central autonomic nervous system (CANS). To clarify the central part in the autonomic nervous system more clearly, we have corrected the statement “central autonomic nervous system” to “central autonomic pathways” in the revised manuscript (p. 4, 5, 13, 44).

1. Ulusoy Ayse., Phillips Robert J., Helwig Michael., Klinkenberg Michael., Powley Terry L., Di Monte Donato A., (2017). Brain-to-stomach transfer of α -synuclein via vagal preganglionic projections., *Acta Neuropathol.*, 133, 381-393.
2. De Groat W C., Booth A M., Milne R J., Roppolo J R., (1982). Parasympathetic preganglionic neurons in the sacral spinal cord., *J. Auton. Nerv. Syst.*, 5, 23-43.
3. Orimo Satoshi., Ghebremedhin Estifanos., Gelpi Ellen., (2018). Peripheral and central autonomic nervous system: does the sympathetic or parasympathetic nervous system bear the brunt of the pathology during the course of sporadic PD?, *Cell Tissue Res.*, 373, 267-286.

2. During the cardiovascular test, the results do not show changes in heart rate. How do the authors explain that, if the heart rate would be the responsible for cardiac output and consequently by the changes in blood pressure? Is possible that the reduction in the blood pressure be a result of a vascular instead of baroreflex dysfunction?

R: These are very good questions. As you mentioned, heart rate is associated with cardiac output and blood pressure. When the blood pressure is too low, the baroreceptors send signals to the central nervous system and then activate the sympathetic nervous system, which stimulates the circulatory system to induce vasoconstriction and cardioacceleration (PMID: 28717053) [1]. The cardiovascular autonomic system with normal structure and function is the foundation of the baroreflex and critical to controlling adequate blood pressure. If any abnormality occurs in this system, the decrease of blood pressure will be unable to cause a rise in heart rate reflectively. In our research, injection of α -Syn PFFs into stellate ganglia leads to pathological α -Syn accumulation in the nerve terminals of both the myocardium and the vessel wall (as shown in Fig. 2 in the manuscript). The deposition of α -Syn aggregates in the cardiovascular autonomic nervous system causes impairment of the autonomic pathways of the myocardium and vessels. Therefore, the abnormality of the cardiac autonomic pathway caused the unchanged heart rate and the abnormality of the vascular autonomic pathway caused the hypotension in our mouse model. Hence, the heart rate exhibited inconsistency with the blood pressure and cardiac output is owing to the impaired cardiovascular autonomic system and the abnormal baroreflex function. Taken together, we consider that deposition of pathological α -Syn in the cardiovascular system leads to cardiovascular autonomic dysfunction, which in turn leads to baroreflex dysfunction. The dysregulation of heart rate and blood pressure is a result of the baroreflex dysfunction. Following your suggestion, we have edited the Discussion section of the revised manuscript (p. 16; lines 336-342).

1. Brown Thomas P., (2017). Pure autonomic failure., *Pract Neurol*, 17, 341-348.

3. *The immunohistochemical experiments should be performed with double staining to conclude which neuronal population was affected. Some examples: TH + Syn for staining C1 and A6 regions; Choline acetyltransferase + Syn for staining nucleus Ambiguus, dorsal motor nucleus of vagus and preganglionic neurons; tryptofan-hydroxylase + Syn for staining raphe.*

R: Your comment here is constructive. Following your suggestion, we have used anti-Choline acetyltransferase (ChAT), tryptophan-hydroxylase (TPH) and tyrosine hydroxylase (TH) antibodies to detect different types of neurons and anti-glial fibrillary acidic protein (GFAP) antibody for astrocytes. The result of double staining shows co-localizations of p α -Syn positive inclusions with TH in A6 cell group of locus coeruleus (a) and C1 cell group of medulla oblongata (b), ChAT in ambiguus nucleus (c) and dorsal motor nucleus of vagus (d), TPH in dorsal raphe nuclei, dorsal part (DRD) (e), and no co-localization with GFAP in pontine reticular nucleus, caudal part. (shown in **Fig. 1** below). We have added the respective images in the new Fig. 2 in the revised manuscript and have made modifications in the Results section (p. 5-6, lines 109-130, p. 33-35, lines 771-782).

Fig. 1. Double immunofluorescence analysis of the pons and medulla oblongata from α -Syn PFFs mice for α -Syn (green) with TH in A6 cell group of locus coeruleus (red, a) and C1 cell group of medulla oblongata (red, b), ChAT in ambiguus nucleus (red, c) and dorsal motor nucleus of vagus (red, d), TPH in dorsal raphe nucleus, dorsal part (red, e), GFAP in pontine reticular nucleus, caudal part (red, f). Co-immunolabeling is represented by signal in yellow. Cell nuclei were counterstained with Hoechst 33258 (blue). [Scale bars, 40 μ m (a, c, d, e); 50 μ m (b, f)].

4. *The authors performed the injections into sympathetic ganglia. So, why is possible to see the presence of Syn in parasympathetic preganglionic neurons as NA and DMV? How can the authors justify the changes in gastrointestinal motility if only the sympathetic pathway is affected?*

R: Thank you for your comments and constructive suggestions. We also considered this question when we analyzed the initial results. Then, we performed the experiment, truncal vagotomy, and explored the possibility of α -Syn PFFs spreading from the sympathetic neurons to the parasympathetic ones. To evaluate whether pathological α -Syn spreading via the vagal nerve is required for the pathological deficits in the dorsal motor nucleus of the vagus in α -Syn PFFs-injected mice, we cut off the vagal nerve distributed near the gastrointestinal tract and injected α -Syn PFFs to celiac ganglia 3 days later (shown in **Fig. 2** below). We found that there is no α -Syn inclusion in the dorsal motor nucleus of vagus in α -Syn PFFs-injected mice in which a truncal vagotomy was performed at 7 mpi. However, these mice still displayed significant α -Syn pathology in other regions and nuclei as early as 2 mpi, as shown in **Fig. 2** below. Therefore, we conclude that the presence of pathological α -Syn in

parasympathetic preganglionic neurons as NA and DMV is a result of the pathological α -Syn retrogradely spreading from the vagal nerve terminals. We infer that the pathological α -Syn from the sympathetic ganglia could spread to the heart, blood vessels, and gastrointestinal tract where it is taken up by the vagal nerve terminals and further spreads to NA and DMV transsynaptically and retrogradely. The above findings indicate that exogenous α -Syn PFFs injected into stellate and celiac ganglia spread through both the sympathetic pathway (via the IML) and the parasympathetic pathway (via the vagal nerve).

Similar phenomenon has been demonstrated in a rat model of α -Syn PFFs spreading (PMID: 31254094) [1]. We believe that the changes in gastrointestinal motility in the α -Syn PFFs mice are not only related to impaired sympathetic pathways but also to impaired parasympathetic pathways. Following your suggestions, we have included **Fig. 2** below as a new Supplementary Fig. 3 (p. 7; lines 90-95) in the revised supplement, and made some necessary modifications in the revised manuscript (p. 16, 17; lines 349-358).

Fig. 2. (A and B) Photography (x40) of topographic anatomy of the vagal nerve before (A) and after truncal vagotomy (B) under the stereo-microscope. (C) Representative immunohistochemical results of medulla oblongata from pathological α -Syn PFFs mice receiving truncal vagotomy. Pathologic α -Syn stained with anti-phospho- α -Syn (Ser 129) antibody. [Scale bar, 500 μ m]. (D) Schematic displaying the truncal vagotomy near the gastrointestinal tract and α -Syn PFFs injection to celiac ganglia.

1. Van Den Berge Nathalie., Ferreira Nelson., Gram Hjalte., Mikkelsen Trine Werenberg., Alstrup Aage Kristian Olsen., Casadei Nicolas., Tsung-Pin Pai., Riess Olaf., Nyengaard Jens Randel., Tamgüney Gültekin., Jensen Poul Henning., Borghammer Per., (2019). Evidence for bidirectional and trans-synaptic parasympathetic and sympathetic propagation of alpha-synuclein in rats., *Acta Neuropathol.*, 138, 535-550.

5. It is surprising do not see the presence of Syn in C1 RVLM, especially C1 region, since those neurons are presympathetic and involved in the control of vascular resistance and heart rate.

R: Thank you for your careful observation. You are right that neurons in the rostral ventrolateral medulla (RVLM) are presympathetic and involved in the control of vascular resistance and heart rate. Previous studies have also found that the RVLM projects to the intermediolateral nucleus (IML) in the spinal cord (PMID: 27844333) [1]. In fact, we have detected the presence of phosphorylated α -Syn in the RVLM. Figure 1 in the manuscript shows mild phosphorylated α -Syn aggregates in the RVLM. Here, we have attached a clearer picture (**Fig. 3** below) to show the pathological α -Syn accumulated in the RVLM.

Fig. 3. Immunohistochemical results of α -Syn pathology in the medulla oblongata (A) and rostral ventrolateral medulla (RVLM) (B) from pathological α -Syn PFFs mice. B shows a higher magnification relative to A. [Scale bars, 250 μ m (A), 25 μ m (B)].

1. Jamali Hina K., Waqar Fahad., Gerson Myron C., (2017). Cardiac autonomic innervation., J Nucl Cardiol, 24, 1558-1570.

6. The authors need to provide more information in the Methods section about antibodies, the number of sections analyzed per brain, the thickness of the sections and the distance between the sections analyzed.

R: Thank you for your constructive advice. We have used 17 commercially available antibodies, as shown in **Table 1** below. Serial coronal sections of the medulla, pons, midbrain, and olfactory bulb were analyzed, though we have displayed only 4

representative sections of medulla oblongata (Bregma -6.72 mm), pons (lower, Bregma -5.40 mm), pons (upper, Bregma -4.48 mm), and midbrain (Bregma -2.48 mm) in the manuscript. Immunohistochemistry was performed using paraffin sections of 4 μ m thickness. The distance between medulla oblongata (Bregma -6.72 mm) and pons (lower, Bregma -5.40 mm) is about 1.32 mm, between pons (lower, Bregma -5.40 mm) and pons (upper, Bregma -4.48 mm) is about 0.92 mm, and between pons (upper, Bregma -4.48 mm) and midbrain (Bregma -2.48 mm) is about 2.00 mm. Following your advice, we have included this table as a new Supplementary Table 1 (p. 8) and provided more detailed information in the Methods section of the revised manuscript (p. 21, lines 449-455).

Table 1. Antibodies used in the study

Antibodies	Source/Cat.No./precise use name	Host	Dilution	Detailed feature
Anti-phospho- α -Syn (129)	(Ser Millipore (MABN826) (81A)	Mouse	1:600 (IHC) 1:800 (IF)	Detecting CK1- and CK2-catalyzed α -Syn Ser129 phosphorylation
Anti-phospho- α -Syn (129)	(Ser Abcam (ab51253) (EP1536Y)	Rabbit	1:600 (IHC) 1:800 (WB)	Detecting α -Syn phosphorylated on Ser129
Anti-phospho- α -Syn (129)	(Ser Wako (015-25191) (pSyn#64)	Mouse	1:500 (IHC) 1:1000 (WB)	Detecting α -Syn including phosphorylated Ser129
Anti-Ubiquitin	Cell Signaling Technology (3933)	Rabbit	1:500 (IHC, IF)	----
Anti-Ubiquitin	Millipore (MAB1510)	Mouse	1:500 (IF)	----
Anti-Glial fibrillary acidic protein	Abcam (ab4674)	Chicken	1:800 (IF)	----
Anti-Microtubule-associated protein-2	Abcam (ab32454)	Rabbit	1:800 (IF)	----
Anti-Myelin Basic Protein	Abcam (ab40390)	Rabbit	1:900 (IF)	----
Anti-Tyrosine Hydroxylase	Abcam (ab112)	Rabbit	1:800 (IHC)	----
Anti-Tryptophan Hydroxylase	Abcam(ab52954)	Rabbit	1:800 (IHC)	----
Anti-Choline Acetyltransferase	Abcam(ab18736)	Sheep	1:800 (IHC)	----
Anti-Glyceraldehyde-3-Phosphate Dehydrogenase	Millipore (MAB374)	Mouse	1:1000 (WB)	----
Cy TM 2 AffiniPure Donkey Anti-Rabbit IgG (H+L)	Jackson ImmunoResearch (711-225-152)	Donkey	1:400 (IF)	----
Rhodamine Red TM -X (RRX) AffiniPure Donkey Anti-Mouse IgG (H+L)	Jackson ImmunoResearch (715-295-151)	Donkey	1:400 (IF)	----
Cy TM 2 AffiniPure Donkey Anti-Chicken IgG (H+L)	Jackson ImmunoResearch (703-225-156)	Donkey	1:400 (IF)	----
Anti-Mouse IgG (H+L) HRP Conjugate	Promega (W4021)	Goat	1:2500 (WB)	----

Minor:

1. I would expect some kind of progressive functional responses. It looks that as soon as the dysautonomia appears, it is total.

R: Thank you for your careful observation and good suggestions for improving our manuscript. We have also considered this issue, therefore, we tried to inject α -Syn PFFs into stellate ganglia and celiac ganglia of TgM83^{+/-} mice, separately. The survival curves of these two groups of mice are shown in **Fig. 4** below. The median survival times of the α -Syn PFFs mice injected in stellate ganglia (SG) alone and celiac ganglia (CG) alone are 28 weeks and 38 weeks of age, respectively. Thus, the α -Syn PFFs mice injected in SG alone showed diseased phenotypes earlier than the mice injected in CG alone. This provides evidence that the involvement of the cardiovascular system accounts for the rapid death of mice to a large extent. The rapid spread of α -Syn to vital organs for life, such as the heart and blood vessels, potentially causes death of α -Syn PFFs mice once the phenotypes appear. Additionally, the time points we have used may not cover all of the changing points to functional responses. This reminds us that we can take more detailed points for research in future experiments. Thank you again for your suggestion.

Fig. 4. Kaplan–Meier survival plot shows survival time (weeks of age) of α -Syn PFFs mice and PBS mice injected in SG alone and CG alone. n = 10 per group.

2. The authors perform injections into the ganglion. However, it is not clear how the authors know that they reach the ganglion.

R: At the beginning of our experiment, we conducted a series of pre-tests to confirm the successful injections into the stellate ganglia or celiac ganglia. Prior to injection of PFFs, trypan blue was injected into the ganglia to determine the volume that could be injected without substantial damage or leakage as shown in **Fig. 5** below. Based on these experiments, it was determined that 1.5 μ l for the stellate ganglia and 4.0 μ l for

the celiac ganglia could be safely injected under the stereo-microscope. We also added some descriptions to the Methods section of the revised manuscript (p. 20; lines 427-431) and included these pictures as new Supplementary Fig. 1A (p. 2, 3; lines 39, 40).

Fig. 5. Photography (x60) of a stellate (left panel) and celiac (right panel) ganglion after trypan blue injection under the electronic microscope.

3. It would be interesting to describe the advantages and disadvantages of this model

R: Thanks for your advice. Here we list the advantages and disadvantages of this model. Advantages: In the present study, we have shown that inoculation of α -Syn PFF into stellate and celiac ganglia induces spreading of α -Syn pathology through the autonomic pathway to both the central nervous system and the autonomic innervation of peripheral organs bidirectionally in TgM83^{+/-} mice. Parallely, the mice develop various autonomic dysfunctions without motor dysfunction. Therefore, we generated a mouse model with pure autonomic dysfunction caused by α -Syn pathology *in vivo*. This mouse model may help define the mechanistic link between the transmission of pathological α -Syn and the cardinal features of autonomic dysfunction in α -synucleinopathies. Disadvantages: Firstly, the α -Syn-injected mice died within a short time once the phenotypes appear and the time points we chose may not cover all of the changing points to functional responses. So, we fail to observe the progressive functional responses. Secondly, in fact, the wild-type (WT) mice have a pure background, and are more helpful for explaining the scientific issues. In the following study, maybe we could try to inject the mouse α -Syn PFFs into the ganglia of WT mice, and continue to observe the pathological and behavioral changes. To improve the quality of the manuscript, we have added the relevant statement in the Discussion section of the revised manuscript (p. 17, 18; lines 376-381).

4. Figure 2: which region of the medulla was analyzed?

R: Per your advice, we have adjusted the figure (Fig. 2 in the revised manuscript, p.

33-35, lines 771-782). We used tyrosine hydroxylase (TH), choline acetyltransferase (ChAT), tryptophan hydroxylase (TPH) and microtubule-associated protein-2 (MAP-2) for neurons; glial fibrillary acidic protein (GFAP) for astrocytes; and myelin basic protein (MBP) for oligodendrocytes; as well as ubiquitin (Ub) for inclusions. Double IF results revealed the co-localizations of p α -Syn inclusions with TH in the A6 cell group of the locus coeruleus (LC) and C1 cell group of the medulla oblongata, ChAT in the ambiguus nucleus (Amb) and dorsal motor nucleus of vagus (10N), TPH in the dorsal raphe nucleus, dorsal part (DRD), MAP-2 in the parvicellular reticular nucleus, alpha part (PCRtA). Neurons contain the most abundant p α -Syn-positive inclusions, while no appreciable co-localization of p α -Syn with GFAP or MBP in the pons or the medulla oblongata were observed. P α -Syn inclusions are positive for Ub in the medulla oblongata and pons.

5. Figure 4K: which is the staining for TH? Where is the Syn?

R: TH-positive staining is visualized by 3,3'-diaminobenzidine (DAB) staining, as shown in **Fig. 6** below, with arrows indicating the TH-positive staining. Fig. 4K in the manuscript shows that the TH-positive soma was obviously reduced in the olfactory bulb (Ob) from pathological α -Syn PFFs mice compared to PBS mice. We have also performed immunohistochemistry analysis using anti-phospho- α -Syn (Ser 129) antibody, anti-aggregated α -Syn antibody, and anti- α -Syn filament. However, little pathological α -Syn was detected in the Ob from both pathological α -Syn PFFs mice and PBS mice. As previously reported, GABAergic inhibitory interneurons play a central role in the encoding of olfactory information (PMID: 25216299) [1]. And virtually all (97%) TH-positive cells are DAergic-GABAergic neurons (PMID: 20089927) [2]. Therefore, we speculate that the reduction of TH-positive cells is the reason for the olfactory deficits in pathological α -Syn PFFs mice.

Fig. 6. Tyrosine hydroxylase (TH) staining in olfactory bulb (Ob) of pathological α -Syn PFFs mice. The arrows show the TH-positive staining.

1. Bonzano Sara., Bovetti Serena., Fasolo Aldo., Peretto Paolo., De Marchis Silvia., (2014). Odour enrichment increases adult-born dopaminergic neurons in the mouse olfactory bulb., *Eur. J. Neurosci.*, 40, 3450-7.
2. Kiyokage Emi., Pan Yu-Zhen., Shao Zuoyi., Kobayashi Kazuto., Szabo Gabor., Yanagawa Yuchio.,

Obata Kunihiko., Okano Hideyuki., Toida Kazunori., Puche Adam C., Shipley Michael T., (2010). Molecular identity of periglomerular and short axon cells., J. Neurosci., 30, 1185-96.

6. *Why do the animals that receive Syn injection die earlier?*

R: Thank you for your question. α -Syn PFFs were injected into the stellate ganglia and celiac ganglia of mice, which damaged the autonomic system. With the widespread α -Syn pathology in the central autonomic pathways and peripheral autonomic system, the α -Syn PFFs mice developed a series of autonomic phenotypes (orthostatic hypotension, constipation and hypohidrosis) and olfactory dysfunction gradually. In particular, the rapid spread of pathological α -Syn affects vital organs for life such as the heart and blood vessels, which potentially results in the death of the α -Syn PFFs mice within a short time once the phenotypes appear. Therefore, PFFs-injected mice die earlier than PBS-injected mice.

7. *How was the drink and food intake?*

R: Thank you for your careful observation and we apologize for the omissions. Each mouse was given 3 to 7 g/day of feed, and 4 to 7 ml/day of purified water. The mice were given free access to feed and water. We have added a related description to the Methods section of the revised manuscript (p. 18; line 395).

REVIEWER #2

Xue-Jing Wang et al. present an interesting story about autonomic neuropathology and phenotype induced by autonomic ganglionic injection of a-syn PFFs. The idea is novel, and the quality of majority data looks nice. Some experiment design and rationale are quite weak, which significantly hamper the enthusiasm on this manuscript. Some experimental details are missing, but are really big concerns for the significance of the manuscript. I would suggest the authors need to revise the manuscript carefully with great effort.

The main weakness of the manuscript is to use the a-syn transgenic mice, which is limited to familial Parkinson's disease. The TgM83 mice start to die from 7-month old, however the experiment design is to sacrifice the mice 7-months after the injection, which has been 9 months old (2-month old receive PFFs injection). The variation of TgM83 in the survival curve may have interfered the conclusion. TgM83 lines express both human A53T and endogenous mouse a-synuclein, and A53T mutant a-syn and mouse a-syn definitely are distinct strain from inoculated human wildtype a-syn PFFs (S-1001, rpeptide). Furthermore, it is not clear that the autonomic dysfunction is caused by the inoculation of PFFs into stellate and celiac ganglia, since aged a-syn transgenic mice may have some non-motor symptoms and even pathology in the gut as well (PMID: 20106867). What about aged TgM83, do they have the some similar autonomic dysfunction or phenotype?

Supplementary Fig. 1B showed the size of a-syn PFFs is very long (> 1000 nm), which is quite different from the size (< 100 nm) in the published papers, please see PMID: 23161999, 27708076. There are some other lowercase letters, which are confusing.

Supplementary Fig. 2A, the mouse number is missing, but based on the dots in the plot, there are only 5 mice for each group, although the two groups may exhibit dramatic difference in the survival curve. Fig. 2B, how many mice are included in the result, and what is the error bar? How well these non-motor syndrome can be characterized in mouse model? Is there any well-established paper for citation? Is there any correlation between the non-motor symptoms appeared in different timeline and the alpha-synuclein pathology appeared in different timeline? Does the observed pathology pattern fit in the PAF?

Fig. 2C, the quality of the pathology is too low, high magnified images should be provided.

Some references should be cited, including LAG3 as the receptor of a-syn PFFs (PMID: 27708076), gut-PFFs model (Neuron 2019 in press).

The hypohidrosis results are related to number of paws or number of animal?

Since the experiment design contains several uncertain factors, which significantly reduce the quality of the discovery, and hamper us to effectively studying the role of a-syn PFFs in a-syn spreading from the stellate and celiac ganglia.

All the histogram should be shown with scattered dots, to understand the distribution of the raw data. Power analysis is missing. All the experiments should be repeated with three independent times. Mice number in each experiment may need > 3 mice at least; 3 mice may be good if power analysis results show the reasonability.

Fig.2, why there are some overlap with GFAP and pa-syn, is it the phenotype of PAF? Fig.3, quantification is missing. Big scale of images should be shown to understand the distribution.

What is site of the phosphorylated a-syn or pa-syn, which is not described clearly in the manuscript?

All the symptoms including hyposmia, hypohidrosis, constipation, and orthostatic hypotension is the non-motor symptoms of Parkinson's disease (PD), however, the authors failed to show the motor-related symptoms as well as the loss of dopaminergic, which may be caused by short incubation time due to TgM83 mice (see WT mice in Neuron 2019 in press)? This is a weakness, or may suggest that it is not a good disease model? What about dementia?

What about the endotoxin, which has been suggested playing an important role in the a-syn strain formation and the pathogenesis of a-syn spreading? The symptoms observed in the manuscript could be attributed by the endotoxin. The good control is a-syn monomer.

Response to REVIEWER #2

R: Thank you for your time and constructive suggestions for improving our manuscript. We have carefully studied your comments and addressed them in the following parts.

1. The main weakness of the manuscript is to use the α -syn transgenic mice, which is limited to familial Parkinson's disease. The TgM83 mice start to die from 7-month old, however the experiment design is to sacrifice the mice 7-months after the injection, which has been 9 months old (2-month old receive PFFs injection). The variation of TgM83 in the survival curve may have interfered the conclusion. TgM83 lines express both human A53T and endogenous mouse α -synuclein, and A53T mutant α -syn and mouse α -syn definitely are distinct strain from inoculated human wildtype α -syn PFFs (S-1001, rpeptide). Furthermore, it is not clear that the autonomic dysfunction is caused by the inoculation of PFFs into stellate and celiac ganglia, since aged α -syn transgenic mice may have some non-motor symptoms and even pathology in the gut as well (PMID:20106867). What about aged TgM83, do they have the some similar autonomic dysfunction or phenotype?

R: Thank you for your comments. As you mentioned, the TgM83 mouse model is a classical model for familial Parkinson's disease, and they express both the mouse endogenous α -Syn and the human α -Syn-A53T (PMID: 12062037) [1]. Mouse and human α -Syn may have some differences in the induction of α -synucleinopathy models according to previous literatures. An excellent research published in the *Neuron* this year (PMID: 31255487) [2] found that the mouse α -Syn PFFs-injected mice showed pathological α -Syn accumulation in the central nervous system at 3 months after injections into the pyloric stomach and upper duodenum of C57BL/6J mice. However, the human α -Syn PFFs-injected mice failed to elicit pathological α -Syn accumulation at the same time point. This study has indicated to us that the mouse α -Syn PFFs have the absolute advantage in inducing pathological α -Syn accumulation in mice. Accordingly, we suppose that human α -Syn PFFs are prone to induce the spreading of pathological human α -Syn. TgM83 mice express human α -Syn-A53T transgene, whose expression level was much higher than that of the mouse endogenous α -Syn. We chose the TgM83 model and tried to induce the formation and spreading of human α -Syn inclusions along the autonomic pathway through the injection of human α -Syn PFFs. We also injected human α -Syn PFFs into the stellate ganglia and celiac ganglia of C57BL/6 wildtype mice, and did not detect spreading of α -Syn (shown in **Fig. 7** below). Our findings also support their research.

As you noted, our model does have some weaknesses. The human α -Syn expressed in TgM83 mice is a mutant protein which facilitates the formation and spreading of pathological α -Syn inclusions after injection in the autonomic nervous system, which is different from the real pathological process of PAF. We have included **Fig. 7** below in the new Supplementary Fig. 2 (p. 4).

According to our study and previous reports (PMID: 12062037) [1], the uninoculated TgM83^{+/-} mice develop motor phenotypes at 22-28 months of age and then die within 2-3 weeks of phenotypic onset. The uninoculated TgM83^{+/+} mice develop similar phenotypes at 8-16 months of age. The median survival time of uninoculated TgM83^{+/+} mice is 14.5 months old (PMID: 21813214) [3].

Thank you for your careful observation and we are sorry for this editorial mistake. We did not indicate the times clearly in the statistical graphs, which may cause potential confusion. In fact, the '7 months' refers to 7 months post-injection in the survival curve rather than the age of mice. In the analyses of autonomic function and olfaction, the time points in the abscissas also refer to the post-injection times. The median survival times of α -Syn PFFs mice and PBS mice injected bilaterally are 24 weeks and 100 weeks of age, respectively. They developed orthostatic hypotension, constipation, hypohidrosis, and hyposmia at 12 weeks post-injection (wpi), 14 wpi, 14 wpi, and 16 wpi, respectively. The median survival times of α -Syn PFFs mice and PBS mice injected unilaterally are 28 weeks and 100 weeks of age, respectively. Thank you for your reminder; the abscissa in the survival curve graph should apply to the survival age of mice instead of the post-injection time points to disambiguate readers. Per your advice, we now use the age of survival as the abscissa of the survival curve (shown in **Fig. 8** below) and clearly mark the abscissa of the autonomic function evaluation chart in Fig. 4 of the revised manuscript (p. 38, 39; lines 822, 823) and Supplementary Fig. 2 of the revised supplement (p. 4, 5; lines 73-75).

We have searched for previous studies and found little description on the autonomic function of TgM83 mice. A study in 2011 reported that TgM83 mice presented urinary dysfunction preceding motoric changes (PMID: 21916020) [4]. In our mouse model, the urodynamic function evaluation of the α -Syn PFFs mice and PBS mice show no abnormality even at the time of death, as shown in **Fig. 9** below. No published research has been found involving other autonomic functions of TgM83 mice such as cardiovascular, gastrointestinal, or sweating function.

Fig. 7. Representative immunohistochemical results of different CNS segments from α -Syn

PFFs-injected C57BL/6 wildtype mice using anti-phospho- α -Syn (Ser 129) antibody. (A-F) The immunohistochemical images display segments including the stellate ganglia (A), celiac ganglia (B), the 3rd thoracic spinal cord (T3) (C), medulla (D), pons (lower) (E), and pons (upper) (F). [Scale bars, 40 μ m (A and B); 200 μ m (C); 500 μ m (D-F)].

Fig. 8. Kaplan–Meier survival plots show survival times (weeks of age) of α -Syn PFFs mice and PBS mice injected bilaterally (A) and unilaterally (B).

Fig. 9. Urinary function analyses of pathological α -Syn PFFs mice and PBS mice at 6 mpi. (A1 and A2) Representative cystometry traces in α -Syn PFFs (A1) and PBS (A2) mice at 6 mpi. (B1-5) Summary bar graphs from urodynamic evaluation for α -Syn PFFs (n =10) and PBS (n =10) mice at 6 mpi including amplitude (B1), #NVCs/Cycle (B2), PVR (B3), VV (B4) and ICI (B5). Data are the means \pm SD. Statistics were analyzed employing the Student's t test and Mann-Whitney test. *P < 0.05 indicates a significant difference between α -Syn PFFs groups and PBS groups. Abbreviations: NVCs: Nonvoiding contractions during filling phase; PVR: Postvoid Residual Volume; VV: Voided volume; ICI: Intercontraction Interval.

1. Giasson Benoit I., Duda John E., Quinn Shawn M., Zhang Bin., Trojanowski John Q., Lee Virginia M-Y., (2002). Neuronal alpha-synucleinopathy with severe movement disorder in mice expressing A53T human alpha-synuclein., *Neuron*, 34, 521-33.
2. Kim Sangjune., Kwon Seung-Hwan., Kam Tae-In., Panicker Nikhil., Karuppagounder Senthilkumar S., Lee Saebom., Lee Jun Hee., Kim Wonjoong Richard., Kook Minjee., Foss Catherine A., Shen Chentian., Lee Hojae., Kulkarni Subhash., Pasricha Pankaj J., Lee Gabsang., Pomper Martin G., Dawson Valina L., Dawson Ted M., Ko Han Seok., (2019). Transneuronal Propagation of Pathologic α -Synuclein from the Gut to the Brain Models Parkinson's Disease., *Neuron*, 103, 627-641.e7.
3. Mougnot Anne-Laure., Nicot Simon., Bencsik Anna., Morignat Eric., Verchère Jérémy., Lakhdar Latefa., Legastelois Stéphane., Baron Thierry., (2012). Prion-like acceleration of a synucleinopathy in a transgenic mouse model., *Neurobiol. Aging*, 33, 2225-8.
4. Hamill Robert W., Tompkins John D., Girard Beatrice M., Kershen Richard T., Parsons Rodney L., Vizzard Margaret A., (2012). Autonomic dysfunction and plasticity in micturition reflexes in human α -synuclein mice., *Dev Neurobiol*, 72, 918-36.

2. Supplementary Fig. 1B showed the size of α -syn PFFs is very long (> 1000 nm), which is quite different from the size (< 100 nm) in the published papers, please see PMID: 23161999, 27708076. There are some other lowercase letters, which are confusing.

R: Thanks for your careful observation and good suggestions. We have calculated the size of more than 500 sonicated α -Syn PFFs and implemented statistical analysis. The average length of sonicated α -Syn PFFs is 47.8 \pm 23.7 nm (shown in **Fig. 10** below). Following the references you noted (PMID: 23161999, 27708076) [1, 2], we have added descriptions in the revised manuscript (p. 4, line 91; p. 26; lines 565-567) and inserted some new images with higher magnification (shown in **Fig. 10** below) in the new Supplementary Fig. 1B and D (p. 2, 3; lines 42-45, 47-50).

Fig. 10. (A) Representative negative-stained transmission electron micrographs of α -Syn PFFs before sonication (a) and after sonication (b). [Scale bars, 100 nm]. (B) Histogram representation of >500 sonicated fibrils measured from randomly captured electron microscopy images. Black bars represent group median, with mean and corresponding group standard deviation indicated in bold font.

1. Luk Kelvin C., Kehm Victoria., Carroll Jenna., Zhang Bin., O'Brien Patrick., Trojanowski John Q., Lee Virginia M-Y., (2012). Pathological α -synuclein transmission initiates Parkinson-like neurodegeneration in nontransgenic mice., *Science*, 338, 949-53.
2. Mao Xiaobo., Ou Michael Tianhao., Karuppagounder Senthilkumar S., Kam Tae-In., Yin Xiling., Xiong Yulan., Ge Preston., Umanah George Essien., Brahmachari Saurav., Shin Joo-Ho., Kang Ho Chul., Zhang Jianmin., Xu Jinchong., Chen Rong., Park Hyejin., Andrabi Shaida A., Kang Sung Ung., Gonçalves Rafaella Araújo., Liang Yu., Zhang Shu., Qi Chen., Lam Sharon., Keiler James A., Tyson Joel., Kim Donghoon., Panicker Nikhil., Yun Seung Pil., Workman Creg J., Vignali Dario A A., Dawson Valina L., Ko Han Seok., Dawson Ted M., (2016). Pathological α -synuclein transmission initiated by binding lymphocyte-activation gene 3., *Science*, 353, undefined.

3. Supplementary Fig. 2A, the mouse number is missing, but based on the dots in the plot, there are only 5 mice for each group, although the two groups may exhibit dramatic difference in the survival curve. Fig. 2B, how many mice are included in the result, and what is the error bar? How well these non-motor syndrome can be characterized in mouse model? Is there any well-established paper for citation? Is there any correlation between the non-motor symptoms appeared in different timeline and the alpha-synuclein pathology appeared in different timeline? Does the observed pathology pattern fit in the PAF?

R: We really appreciate your careful observation and thoughtful suggestions. In Supplementary Fig. 2A, we missed the number of mice in the figure legend of the survival curve, and we have now added it to the figure legend of Supplementary Fig. 2 in the revised supplement (p. 5; line 74). In fact, the number of unilaterally injected mice is 10 per group. In Fig. 2B, 104 mice are included in the results. In order to display the error bars, we have modified the original image to a bar chart with error bars and scattered dots, as shown in **Fig. 11** below, which has also been placed in Fig. 4B in the revised manuscript (p. 38) and the new Supplementary Fig. 2B (p. 4).

In our experiments, we evaluated the non-motor symptoms of mice with reference to previous literature. We evaluated the cardiovascular function of mice, refer to the literatures (PMID: 24734888, 26188021, 16138782) [1-3], gastrointestinal function, refer to (PMID: 24604453) [4], sweating function, refer to (PMID: 28648603) [5], and olfactory function, refer to (PMID: 25177842, 24362016, 24204848, 25799501) [6-9].

According to the results, we consider that there is a correlation between non-motor symptoms and α -Syn pathology in different timelines in our mouse model. The appearance of α -Syn aggregated inclusions parallels the onset of the non-motor symptoms. As the pathological deficits become more robust, the symptoms of the diseased mice are more severe. We have modified the Discussion section in the revised manuscript, as can be seen in p. 14, lines 303, 304.

In patients with PAF, α -Syn pathology was observed in sympathetic ganglia, IML of the spinal cord, 10N, locus coeruleus (LC), substantia nigra, raphe nucleus, and brainstem reticular formation, nucleus basalis of Meynert, Edinger-Westphal nucleus in CNS. Deposits of phosphorylated α -Syn could also be detected in nerve fibers of skin and gut in PAF patients (PMID: 8307061, 8006660, 9255396, 11109008, 11294945, 16606777) [10-15]. In our mouse models, we detected α -Syn deposits in the sympathetic ganglia, IML, LC, reticular formation, arcuate hypothalamic nucleus (Arc), periaqueductal gray (PAG), ambiguous nucleus (Amb), nucleus of the solitary tract (Sol), and peripheral nerve terminals, which is in accord with the phenotypes of the α -Syn PFFs mice. Therefore, we believe that our model recapitulates the pathological changes of PAF patients to a large extent. Here, we list **Table 2** below, regarding the overview of postmortem studies of PAF patients and pathological α -Syn distribution of the α -Syn PFFs mice in the present study. We have included this table as a new Supplementary Table 2 (p. 9) and edited modifications in the Discussion section of the revised manuscript (p. 17, lines 371-373).

Table 2. Overview of postmortem studies of PAF patients and pathological α -Syn distribution of the α -Syn PFFs mice in the present study

	PAF patients	α -Syn PFFs mice
Similarities	sympathetic ganglia	
	intermediolateral cell column	
	locus coeruleus	
	raphe nucleus	

	dorsal motor nucleus of vagus (10N)	
	periaqueductal gray	
	brainstem reticular formation	
	skin	
	gastrointestinal tract	
	epicardium	
Differences	substantia nigra	arcuate nucleus
	Edinger-Westphal nucleus	nucleus ambiguus
	nucleus basalis of Meynert	nucleus of solitary tract

Fig. 11. Onset time of orthostatic hypotension, constipation, hypohidrosis, and hyposmia in pathological α -Syn PFFs mice injected bilaterally (a) and unilaterally (b).

1. Tanimoto Keiji., Kanafusa Sumiyu., Ushiki Aki., Matsuzaki Hitomi., Ishida Junji., Sugiyama Fumihiko., Fukamizu Akiyoshi., (2014). A mouse renin distal enhancer is essential for blood pressure homeostasis in BAC-rescued renin-null mutant mice., *J. Recept. Signal Transduct. Res.*, 34, 401-9.
2. Regan Jessica A., Mauro Adolfo Gabriele., Carbone Salvatore., Marchetti Carlo., Gill Rabia., Mezzaroma Eleonora., Valle Raleigh Juan., Salloum Fadi N., Van Tassell Benjamin W., Abbate Antonio., Toldo Stefano., (2015). A mouse model of heart failure with preserved ejection fraction due to chronic infusion of a low subpressor dose of angiotensin II., *Am. J. Physiol. Heart Circ. Physiol.*, 309, H771-8.
3. Hagaman John R., John Simon., Xu Lonquan., Smithies Oliver., Maeda Nobuyo., (2005). An improved technique for tail-cuff blood pressure measurements with dark-tailed mice., *Contemp Top Lab Anim Sci*, 44, 43-6.
4. Li Guijie., Wang Qiang., Qian Yu., Zhou Yalin., Wang Rui., Zhao Xin., (2014). Component analysis of Pu-erh and its anti-constipation effects., *Mol Med Rep*, 9, 2003-9.
5. Chee Min Keun., Jo Seong Kyeong., Sohn Kyung Cheol., Kim Chang Deok., Lee Jeung-Hoon., Lee Young Ho., (2017). Effects of Brn2 overexpression on eccrine sweat gland development in the mouse paw., *Biochem. Biophys. Res. Commun.*, 490, 901-905.

6. Lehmkuhl Andrew M., Dirr Emily R., Fleming Sheila M., (2014). Olfactory assays for mouse models of neurodegenerative disease., *J Vis Exp*, undefined, e51804.
7. Ngwa Hilary Afeseh., Kanthasamy Arthi., Jin Huajun., Anantharam Vellareddy., Kanthasamy Anumantha G., (2014). Vanadium exposure induces olfactory dysfunction in an animal model of metal neurotoxicity., *Neurotoxicology*, 43, 73-81.
8. Kurtenbach Stefan., Wewering Sonja., Hatt Hanns., Neuhaus Eva M., Lübbert Hermann., (2013). Olfaction in three genetic and two MPTP-induced Parkinson's disease mouse models., *PLoS ONE*, 8, e77509.
9. Zhang Sufang., Xiao Qian., Le Weidong., (2015). Olfactory dysfunction and neurotransmitter disturbance in olfactory bulb of transgenic mice expressing human A53T mutant α -synuclein., *PLoS ONE*, 10, e0119928.
10. Terao Y., Takeda K., Sakuta M., Nemoto T., Takemura T., Kawai M., (1993). Pure progressive autonomic failure: a clinicopathological study., *Eur. Neurol.*, 33, 409-15.
11. van Ingelghem E., van Zandijcke M., Lammens M., (1994). Pure autonomic failure: a new case with clinical, biochemical, and necropsy data., *J. Neurol. Neurosurg. Psychiatry*, 57, 745-7.
12. Hague K., Lento P., Morgello S., Caro S., Kaufmann H., (1997). The distribution of Lewy bodies in pure autonomic failure: autopsy findings and review of the literature., *Acta Neuropathol.*, 94, 192-6.
13. Arai K., Kato N., Kashiwado K., Hattori T., (2000). Pure autonomic failure in association with human alpha-synucleinopathy., *Neurosci. Lett.*, 296, 171-3.
14. Kaufmann H., Hague K., Perl D., (2001). Accumulation of alpha-synuclein in autonomic nerves in pure autonomic failure., *Neurology*, 56, 980-1.
15. Compta Yaroslau., Martí Maria José., Paredes Pilar., Tolosa Eduardo., (2006). Pure autonomic failure with altered dopamine transporter imaging., *Arch. Neurol.*, 63, 604-5.

4. Fig. 2C, the quality of the pathology is too low, high magnified images should be provided.

R: Thank you for your suggestions. We have replaced the image with a higher magnified picture (Fig. 3C) in the revised manuscript (p. 36, 37).

5. Some references should be cited, including LAG3 as the receptor of a-syn PFFs (PMID: 27708076), gut-PFFs model (Neuron 2019 in press).

R: We appreciate your suggestion and have cited additional related references, including the two papers from Dr. T Dawson's group in our revised manuscript. Also, we have added related statements in the Discussion section of the revised manuscript (p. 13; lines 281-283, 287-289).

6. The hypohidrosis results are related to number of paws or number of animal?

R: Thank you for your question. We express our hypohidrosis results as number of spots per paw as opposed to number of spots per animal. The starch-iodine assay is utilized to evaluate sweating function in the mouse model. The assay reflects sweating

spots of mice after pilocarpine injection in a straightforward manner. The sweating spots in the paw is a main indicator of sweating function (PMID: 25329695, 28347916, 8608813) [1-3]. The sweat response 3 min after pilocarpine injection in the mouse paws was detected by amylase reaction with iodine/starch. Sweating spots in the paw were photographed by digital camera. Then, we used the Imagepro plus to calculate the number of spots in a paw. It is a practical and convenient method to assess sweating function (PMID: 28648603) [4].

1. Klar Joakim., Hisatsune Chihiro., Baig Shahid M., Tariq Muhammad., Johansson Anna C V., Rasool Mahmood., Malik Naveed Altaf., Ameer Adam., Sugiura Kotomi., Feuk Lars., Mikoshiba Katsuhiko., Dahl Niklas., (2014). Abolished InsP3R2 function inhibits sweat secretion in both humans and mice., *J. Clin. Invest.*, 124, 4773-80.
2. Wang ZengYong., Wang Qian., Zhang Man., Hu XueYan., Ding GuoYu., Jiang Min., Bai Gang., (2017). Cimidifugamide from Cimidifuga rhizomes functions as a nonselective β -AR agonist for cardiac and sudorific effects., *Biomed. Pharmacother.*, 90, 122-130.
3. Shimazu M., Matsumoto T., Kosaka M., Ohwatari N., Tsuchiya K., Ueyama Y., Urano K., Katakai Y., Saito M., (1996). A new approach to analysis of human sweating., *Experientia*, 52, 131-5.
4. Chee Min Keun., Jo Seong Kyeong., Sohn Kyung Cheol., Kim Chang Deok., Lee Jeung-Hoon., Lee Young Ho., (2017). Effects of Brn2 overexpression on eccrine sweat gland development in the mouse paw., *Biochem. Biophys. Res. Commun.*, 490, 901-905.

7. *Since the experiment design contains several uncertain factors, which significantly reduce the quality of the discovery, and hamper us to effectively studying the role of α -syn PFFs in α -syn spreading from the stellate and celiac ganglia.*

R: We agree with you on this comment. As you mentioned, we were also aware of some uncertain factors in the experiments. In order to reduce the impact of the uncertain factors, we had included several “control” groups at the initial stage of the experiments (as shown in **Table 3** below), including the unilateral injection into the stellate ganglia (SG) plus celiac ganglia (CG). We used PBS-injected TgM83^{+/-} mice and α -Syn monomer-injected TgM83^{+/-} mice as control groups. In addition, we also tried this experiment in C57Bl/6J WT background mice. In the α -Syn PFFs mice unilaterally injected in the SG plus CG, the emergence times of pathological α -Syn accumulation and autonomic dysfunctions and their times of death were later, compared with α -Syn PFFs mice injected bilaterally (Details in the supplement). No pathological change or phenotype was detected in the α -Syn monomer or PBS-injected TgM83^{+/-} mice even at 7 mpi, as shown in **Fig. 12** below. In α -Syn PFFs-injected and PBS-injected C57Bl/6 WT mice, there is no pathological α -Syn or phenotype induced, as shown in **Fig. 12** below.

Overall, these control groups we designed to largely reduce the uncertainty of the experiments. Certainly, in the following study, we could explore the effects of exogenous inoculation of mouse α -Syn PFFs in C57Bl/6J WT mice to further investigate the mechanism of α -Syn PFFs in synucleinopathy. We have included **Fig.**

12 below as a new Supplementary Fig. 2D (p. 4-6, lines 83-88) and added related statements in the Discussion section of the revised manuscript (p. 17, 18; lines 359-363, 379-381).

Table 3. Modeling summary

Groups	Inoculum	TgM83 ^{+/-} mice			C57BL/6 mice
		PFFs/PBS		Monomers	PFFs/PBS
	Position	Bilateral SG+CG	Unilateral SG+CG	Bilateral SG+CG	Bilateral SG+CG
Total number of models	PFFs	104	62	12	10
	PBS	60	20	8	6
Cumulative mortality at 5 mpi	PFFs	84.6%	72.6%	8.3%	0.0%
	PBS	3.3%	5.0%	0.0%	0.0%
Cumulative mortality at 6 mpi	PFFs	91.3%	90.3%	8.3%	0.0%
	PBS	6.7%	5.0%	0.0%	0.0%
Cumulative mortality at 7 mpi	PFFs	100.0%	100.0%	8.3%	10.0%
	PBS	8.3%	15.0%	12.5%	10.0%
Cumulative mortality at 24 mpi	PFFs	100.0%	100.0%	100.0%	60%
	PBS	100.0%	100.0%	100.0%	66.7%

Fig. 12. (A and B) Representative immunohistochemical results of different CNS segments from α -Syn monomer-injected TgM83^{+/-} mice (A) and α -Syn PFFs-injected C57BL/6 wildtype mice (B) using anti-phospho- α -Syn (Ser 129) antibody. The immunohistochemical images display segments including the stellate ganglia (a, g), celiac ganglia (b, h), the 3rd thoracic spinal cord (T3) (c, i), medulla oblongata (d, j), pons (lower) (e, k), and pons (f, l). [Scale Bars, 40 μ m (a, b, g, h); 200 μ m (c, i); 500 μ m (d-f, j-l)].

8. All the histogram should be shown with scattered dots, to understand the distribution of the raw data. Power analysis is missing. All the experiments should be repeated with three independent times. Mice number in each experiment may need > 3 mice at least; 3 mice may be good if power analysis results show the

reasonability.

R: Thank you for your meaningful suggestions that help us improve the quality of our paper. The scatter plot is more accurate, we have carried out further mapping analysis of the relevant results. The histograms in the revised manuscript (p. 38, 41) and supplement (p. 4) have been shown with scattered dots, as seen in **Fig. 13** and **14** below. In the study, we did repeat all the experiments with three independent times. For the bilateral injected TgM83^{+/-} mice in both SG and CG, we used 104 mice to generate different models in the experimental groups. These 104 mice were subjected to autonomic function and motor function every two weeks after modeling until 7mpi. We analyzed data of autonomic function and motor function evaluation from 20 of these mice that were used for the autonomic function and motor function evaluation at 5 mpi, and used the data for statistical analysis and charting. Among 104 α -Syn PFFs mice and 60 PBS mice, 48 α -Syn PFFs mice (8 mice per month after injection until 6 mpi) and 24 PBS mice (4 mice per month after injection until 6 mpi) were subjected to immunohistochemistry and immunofluorescence. Also, 18 α -Syn PFFs mice and 18 PBS mice (3 mice per month after injection until 6 mpi) were subjected to Western blot. We used 20 mice for statistical analysis of survival plot, autonomic function evaluation, and behavioral test. For the unilateral injected TgM83^{+/-} mice in SG and CG, we used 10 mice for statistical analysis of survival plot, autonomic function evaluation, and behavioral test. We have supplemented a power analysis which indicates that the cardiovascular function evaluation, gastrointestinal function evaluation, sweating function evaluation and olfactory function evaluation need 9, 5, 5 and 4 mice, respectively. Following your suggestion, we have included these histograms with scattered dots in new Fig. 4 and Fig. 5 in the revised manuscript (p. 38, 41).

Fig. 13. Onset time (weeks post-injection (wpi)) of constipation, hypohidrosis, and hyposmia

in pathological α -Syn PFFs mice. (A-D) The gastrointestinal transit (A), defecation time of black feces (B) and sweating test in front (C) and hind (D) paws of pathological α -Syn PFFs mice and PBS mice over time. (E-G) Olfaction evaluation. Tyrosine hydroxylase (TH) staining in olfactory bulb (E) of PBS (left) and pathological α -Syn PFFs (right) mice. Bar graphs with scattered dots show the percentage of area occupied by TH immunostaining (F) and the comparison of the buried food test between pathological α -Syn PFFs mice and PBS mice over time (G). $n = 20$ per group. Data are the means \pm SD. Statistical analysis was performed using the Student's t test. **** $p < 0.0001$.

Fig. 14. Behavioral analysis of α -Syn PFFs mice and PBS mice. (A, B) The average time (weeks post-injection (wpi)) to cross the beam (A) and the average number of side slip errors on the beam (B). (C) Latency to fall from the rod. (D) The average time in the hanging wire test. (E, F) Footprint analysis of the fore-base width (E) and the forelimb stride length (F). (G, H) Footprint analysis of the hind-base width (G) and the hindlimb stride length (H). $n = 20$ per group. Data are the means \pm SD. Statistical analysis was performed using the Student's t test, **** $P < 0.0001$.

9. Fig.2, why there are some overlap with GFAP and pa-syn, is it the phenotype of PAF?

R: We used glial fibrillary acidic protein (GFAP) for astrocytes. We did not detect any co-localization of GFAP and $p\alpha$ -Syn. Here we put a high-definition picture (**Fig. 15**

below) for a clearer display. We have searched related literatures carefully and did not find any reports of p α -Syn in astrocytes of pure autonomic failure. Only accumulation of p α -Syn as neuronal cytoplasmic inclusions have been reported in the central nervous system of PAF (PMID: 29297596) [1]. Few postmortem studies of patients with PAF have been reported in the literature, and none have demonstrated astrocyte pathology, although this has been described in other α -synucleinopathies, such as Wakabayashi et al. who demonstrated that α -Syn-positive inclusions were localized in astrocytes of Parkinson's disease brains (PMID: 10651022) [2]. Loria et al. found that α -Syn can be efficiently transferred from astrocytes to astrocytes and from neurons to astrocytes in *in vitro* and *ex vivo* models (PMID: 28725967) [3]. The specific role of astrocytes in α -Syn pathology is not clear and no study has quantitatively assessed the transfer from primary neurons to astrocytes. We think that this may be an interesting and promising subject deserving further exploration in the future.

Fig. 15. Double immunofluorescence analysis of medulla oblongata from α -Syn PFFs mice for GFAP (green) and p α -Syn (red) in dorsal motor nucleus of vagus. Co-immunolabeling is represented by signal in yellow. Cell nuclei were counterstained with Hoechst33258 (blue). [Scale bars, 50 μ m].

1. Coon Elizabeth A., Cutsforth-Gregory Jeremy K., Benarroch Eduardo E., (2018). Neuropathology of autonomic dysfunction in synucleinopathies., *Mov. Disord.*, 33, 349-358.
2. Wakabayashi K., Hayashi S., Yoshimoto M., Kudo H., Takahashi H., (2000). NACP/alpha-synuclein-positive filamentous inclusions in astrocytes and oligodendrocytes of Parkinson's disease brains., *Acta Neuropathol.*, 99, 14-20.
3. Loria Frida., Vargas Jessica Y., Bousset Luc., Syan Sylvie., Salles Audrey., Melki Ronald., Zurzolo Chiara., (2017). α -Synuclein transfer between neurons and astrocytes indicates that astrocytes play a role in degradation rather than in spreading., *Acta Neuropathol.*, 134, 789-808.

10. Fig.2, quantification is missing. Big scale of images should be shown to understand the distribution.

R: Per your advice, we have added images with low magnification (**Fig. 16** below) to show the distribution in the Fig. 2 in the revised manuscript (p. 33).

Fig. 16. Double immunofluorescence analysis of medulla (A) and pons (lower) (B) from α -Syn PFFs mice for p α -Syn (green) and ubiquitin (red). Co-immunolabeling is represented by signal in yellow. [Scale bars, 500 μ m].

11. *What is site of the phosphorylated α -syn or p α -syn, which is not described clearly in the manuscript?*

R: This is a good point. As previously reported, phosphorylation of α -Syn at serine 129 is associated with pathology in α -synucleinopathies (PMID: 27708076) [1]. In our study, the phosphorylated α -Syn or p α -Syn were detected using antibodies against pSer129 α -Syn. Therefore, the phosphorylated α -Syn or p α -Syn mentioned in the manuscript is phosphorylated on Serine129. Some of the modifications can be seen in the revised manuscript (p. 5; line 102).

1. Mao Xiaobo., Ou Michael Tianhao., Karuppagounder Senthilkumar S., Kam Tae-In., Yin Xiling., Xiong Yulan., Ge Preston., Umanah George Essien., Brahmachari Saurav., Shin Joo-Ho., Kang Ho Chul., Zhang Jianmin., Xu Jinchong., Chen Rong., Park Hyejin., Andrabi Shaida A., Kang Sung Ung., Gonçalves Rafaella Araújo., Liang Yu., Zhang Shu., Qi Chen., Lam Sharon., Keiler James A., Tyson Joel., Kim Donghoon., Panicker Nikhil., Yun Seung Pil., Workman Creg J., Vignali Dario A A., Dawson Valina L., Ko Han Seok., Dawson Ted M., (2016). Pathological α -synuclein transmission initiated by binding lymphocyte-activation gene 3., *Science*, 353, undefined.

12. *All the symptoms including hyposmia, hypohidrosis, constipation, and orthostatic hypotension is the non-motor symptoms of Parkinson's disease (PD), however, the authors failed to show the motor-related symptoms as well as the loss of dopaminergic, which may be caused by short incubation time due to TgM83 mice (see WT mice in Neuron 2019 in press)? This is a weakness, or may suggest that it is not a good disease model? What about dementia?*

R: Yes, hyposmia, hypohidrosis, constipation, and orthostatic hypotension are common non-motor symptoms in α -synucleinopathies. There have been many studies related to animal models of PD and motor disorders induced by α -Syn, but few animal

studies focus on non-motor symptoms induced by α -Syn, thus our study attempts to study the non-motor symptoms caused by exogenous injection of α -Syn PFFs. Among α -synucleinopathies, PAF is a pure autonomic disorder without motor symptoms or cognitive deficit, and the pathogenetic mechanism is still unclear.

Our study found that α -Syn pathology selectively attacked central and peripheral autonomic networks instead of the nigrostriatal dopaminergic pathway in the mouse model, accordingly inducing a purely autonomic dysfunction without motor impairment. Hence, our mouse model may be a promising model for investigating autonomic dysfunction induced by pathological α -Syn. For the dementia, we also used the Morris water maze test (MWM) and the Novel object recognition test (NORT) to evaluate the cognitive function in pathological α -Syn PFFs mice at 6mpi. The result shows no evidence of cognitive impairment, as shown in **Fig. 17** below. Nevertheless, a lack of certain phenotypes, such as dopaminergic neuronal loss or dementia, in our study, does not mean that it is not a good disease model. Rather, this unique model provides an opportunity to specifically address autonomic dysfunction in synucleinopathies. In the research field of synucleinopathies, different (genetically modified, or toxins/ α -Syn PFFs-induced) animal models have been reported, it is common that a given model usually recapitulates partial, but not all, phenotypes of a disease, which is in the same case for our models reported here.

Fig. 17. Psychological behavior assessments at 6 months post-injection in pathological α -Syn PFFs mice (n =10) and PBS mice (n =10). (A-E) Representative images of the Morris water maze test (A-C), and the Novel object recognition test (D, E). Statistical analysis was performed using the Student's t test. Error bars represent the mean \pm SEM.

13. What about the endotoxin, which has been suggested playing an important role

in the α -syn strain formation and the pathogenesis of α -syn spreading? The symptoms observed in the manuscript could be attributed by the endotoxin. The good control is α -syn monomer.

R: This is a good point. In the initial stage of the experiments, we had tried to inject α -Syn monomers into the stellate ganglia and celiac ganglia of TgM83^{+/-} mice. No pathological change or phenotype was detected in the α -Syn monomer-injected mice (shown in the answer to your Question 7) in clear contrast to the α -Syn PFFs-injected mice. We therefore believe that the phenotypes observed in the present study are most likely attributed to α -Syn PFFs.

REVIEWER #3

The main objective of the study by Wang et al was to develop a model of pure autonomic failure (PAF) without motor dysfunction as has been the case with other models of alpha-synucleinopathies that target the striato-nigral system. For this purpose, Wang et al inoculated of a-Syn preformed fibrils (PFFs) into stellate of M83 mice a transgenic model expressing low levels of a-Syn. They show that the a-Syn pathology disseminated through the autonomic pathway to the CNS without motor features. The mice also develop functional deficits such as hypotension, constipation and hyposmia. They conclude that their mouse reflects PAF and will help better understand alpha-synucleinopathies with dysautonomia.

This is an interesting study in that this is a relatively new model system of alpha-synucleinopathy with dysautonomia, however several issues need clarification:

1. A number of similar PFF models have been previously developed injecting a-Syn in the periphery developing autonomic deficits. The authors need to clarify in more detail what is different about their model compared to others. It is not completely clear why this model worked differently than other similar models.
2. Need to provide more detailed description and characterization of the PFF used, could it be possible that the animals developed the selective pathology because of a unique strain of a-Syn PFF was used? For example, expand on their EM analysis of the PFF showing if they have a ribbon, helical or other structures similar to what has been shown in models mimicking MSA vs PD. Perform analysis of PK resistance vs degradation or PCMA like analysis.
3. The double labeling image in Figure 2 is very low power and difficult to appreciate the cellular localization of the a-Syn. The patches stained with MBP are unclear what they represent, fiber bundles?, they also need to show double label ubiquitin with cellular marker to understand better the cellular localization, the a-Syn aggregates in red only occasionally appear to co-localize with MAP2 and ubiquitin but nothing else, what is the cellular localization of other aggregates? Are they outside the cells?
4. Did they try this experiment in the WT background? Why there is a need of low levels a-Syn tg to express, this means that their model requires templated seeding with homologous a-Syn (human) and not with mouse a-Syn as it has been shown in the original studies of Luk et al. Is there expression of human a-Syn in the tg model in the periphery, they need to characterize. Are the effects dependent on the expression of the transgene in the autonomic nuclei.
5. It will be important to show that the spreading of the pathology is blocked by blocking the vagal nerve, otherwise the spreading mechanism is unclear given that seeding is occurring in the background or a a-Syn tg model.
6. It would be important to compare to all the relevant controls to better understand to what extent the pathology and functional deficits are the results of synergy between the PFF and the M83 transgene or additive effects or unique to the PFF. The authors only show data on M83 PBS vs PFF, but would be important to compare to WT PBS

vs PFF.

7. Need more detailed statistical analysis and to provide in the figure legends the n number per group. Only males were used, what is the explanation, why they didn't considered sex differences?

Response to REVIEWER #3

R: Many thanks for your precise comments and constructive suggestions for improving our manuscript.

This is an interesting study in that this is a relatively new model system of alpha-synucleinopathy with dysautonomia, however several issues need clarification:

1. A number of similar PFF models have been previously developed injecting α -Syn in the periphery developing autonomic deficits. The authors need to clarify in more detail what is different about their model compared to others. It is not completely clear why this model worked differently than other similar models.

R: We are sincerely grateful for your thoughtful comments. We have thoroughly explored the related literatures, and list a table below (**Table 4**) regarding a general list of rodent models of synucleinopathy induced by exogenous inoculation in which α -Syn PFFs are delivered peripherally.

As **Table 4** shows, rodent models of synucleinopathy induced by exogenous inoculation involve different inocula and inoculation positions, developing variable α -Syn pathology and motor impairments without autonomic dysfunction (details in the table). Therefore, there is a lack of appropriate experimental models to fully recapitulate the entire autonomic dysfunction of α -synucleinopathy without motor impairment. Besides that, the studies of gastrointestinal injection account for the majority of previous studies of peripheral injections with α -Syn PFFs. Our research is based on previous literatures, but different from them in terms of injection position, inoculum, onset time of α -Syn pathology, pathological α -Syn distributed regions, non-motor symptoms, and motor impairments. We injected α -Syn PFFs into the sympathetic ganglia of mice *for the first time*, and the pathological changes mainly existed in the autonomic nervous pathway. Regarding pathological deficits, the α -Syn PFFs mice in our study showed pathological α -Syn accumulation in both CNS and autonomic innervation of peripheral organs as early as 2 months post-injection. Regarding the phenotypes, the α -Syn PFFs mice in our study developed typical and severe autonomic dysfunction (orthostatic hypotension, constipation, and hypohidrosis) and olfactory dysfunction without any motor impairment.

Taken together, we generated a mouse model with pure autonomic dysfunction caused by α -Syn pathology. We hope that this model could help define the mechanistic link between transmission of pathological α -Syn and the cardinal features of autonomic dysfunction in α -synucleinopathy.

Based on your suggestions, we have made some necessary modifications in the Discussion section of the revised manuscript (p.14; line 304-306).

Table 4. List of rodent models of synucleinopathy induced by exogenous inoculation in which α -Syn PFFs are delivered peripherally.

Animal species	Inoculation		Onset time of α -Syn pathology	Pathologic α -Syn distributed regions	Non-motor symptoms	Motor impairments	Reference
	Position	Inoculum					
TgM83 ^{+/+} mice	stellate ganglia + celiac ganglia	α -Syn PFFs	60 days post-injection	SG, CG (+++); Myocardium (++); blood vessels (+); gastrointestinal tract (++); skin (++); IML (+++); Arc (+); LC (+++); PAG (++); Bar (+); RN (++); 10N (++); Amb (+); Sol (+); PnO, PnC, PCRt, Gi (++)	orthostatic hypotension; constipation; hypohidrosis; hyposmia	/	Our study
C57BL/6J mice	the muscle layers of the pylorus and duodenum	mouse α -Syn PFFs	30 days post-injection	dorsal motor nucleus of the vagus, LC, pyloric stomach, upper duodenum, amygdala, substantia nigra pars compacta, hypothalamus, prefrontal cortex, hippocampus, striatum, olfactory bulb	Deficits in Psychological Behavior	Cognitive Deficits	(Sangjune Kim et al., 2019)
Snc α ^{-/-} mice			No pathology	/	/	/	
Homozygous BAC rats	gut wall of the pylorus and duodenum	human α -Syn PFFs or S129A mutant α -Syn PFFs	120 days post-injection	SN pars reticulata, dorsal motor nucleus of the vagus, LC	/	/	(Nathalie Van Den Berge et al., 2019)

Sprague-Dawley rats	left vagus nerve in the neck	human α -Syn -AAVs	180 days post-AAV injection	the dorsal motor nucleus of the vagus, myenteric plexus, nodose ganglion	/	/	(Ayse Ulusoy et al., 2016)
	the right ventral mesencephalon immediately dorsal to the substantia nigra pars compacta			myenteric plexus, myenteric ganglia			
TgM83 ^{+/+} mice	biceps femoris; gastrocnemius muscle	α -Syn PFFs	40-90 days post-injection	Robust in spinal cord, brainstem, and midbrain structures; Sparse in the cortex	/	unilateral foot drop; bilateral foot drop; full hind limb paralysis	(Amanda N. Sacino et al., 2013)
TgM83 ^{+/-} mice	glossal muscles; peritoneal cavity	α -Syn PFFs	Intraperitoneal / intraglossal: 229/285 days postinjection;	Cerebrum (+++); Spinal cord gray matter (+++); Spinal cord ventral horns (++); Cerebellum (-)	/	Weight loss, paralysis, kyphosis, and reduced activity	(Sara Breid et al., 2016)
Sprague Dawley rats	the intestine wall of stomach and duodenum	brain homogenates from PD patients; α -Syn PFFs	48 hours post-injection	Vagal nerve, dorsal motor nucleus of the vagus	/	/	(Staffan Holmqvist et al., 2014)
C57BL/6 J mice	Intraperitoneal cavity; Oral cavity	α -Syn PFFs	No pathology	/	/	/	(Masami Masuda-Suzukake et al., 2014)

The red font represents our study.

2. Need to provide more detailed description and characterization of the PFF used, could it be possible that the animals developed the selective pathology because of a unique strain of α -Syn PFF was used? For example expand on their EM analysis of the PFF showing if they have a ribbon, helical or other structures similar to what has been shown in models mimicking MSA vs PD. Perform analysis of PK resistance vs degradation or PCMA like analysis.

R: These comments are invaluable. We completely agree with you. Baekelandt et. al. speculate that α -Syn misfolded structural variants behave as strains with distinct biochemical and functional properties inducing specific phenotypic traits, which finally provide an explanation for the clinical heterogeneity observed between

Parkinson's disease, MSA, and dementia with Lewy bodies patients (PMID: 26924014) [1]. Moreover, they also show that fibrils seem to be the major toxic strain, resulting in progressive motor impairment and cell death displaying Parkinson's disease and multiple system atrophy traits (PMID: 26061766) [2]. As reported in previous literatures, α -Syn fibrils are widely used in inducing the α -Syn pathology (PMID: 27708076, 25002524, 23161999, 25122523) [3-6]. Moreover, the α -Syn PFFs that we used were generated as previously described (PMID: 27708076) [3]. After sonication, the average size of α -Syn PFFs as determined by transmission electron microscopy was 47.8 ± 23.7 nm (**Fig. 18** below). Furthermore, we have performed the analysis of proteinase K (PK) resistance using a dot-blot test and PK (2.0 μ g/ml) (Solarbio) was used. The α -Syn PFFs we generated are of a fibril nature as revealed by limited proteolysis. As shown for PK digestion in **Fig. 18** below, the α -Syn PFFs resist PK treatment for up to 60 min, however, the monomer is fully degraded. These findings are consistent with the previous report for α -Syn fibrils (PMID: 24108358) [7]. Per your advice, we have included **Fig. 18** below in the new Supplementary Fig. 1 (p. 2, 3, lines 42-50) and added descriptions in the revised manuscript (p. 4, line 91; p. 19, lines 408-410; p. 26, lines 565-567).

Fig. 18. Structural characterization of α -Syn PFFs and monomers. (A) Representative negative-stained transmission electron micrographs of α -Syn PFFs before sonication (a) and after sonication (b). [Scale bars, 100 nm]. (B) Histogram representation of >500 sonicated

fibrils measured from randomly captured electron microscopy images. Black bars represent group median, with mean and corresponding group standard deviation indicated in bold font. (C) A dot-blot reveals degradation patterns of α -Syn monomers (1.0 mg/mL) and α -Syn PFFs (1.0 mg/mL) before and after 60-min incubation in proteinase K.

1. Peelaerts Wouter., Baekelandt Veerle., (2016). α -Synuclein strains and the variable pathologies of synucleinopathies., J. Neurochem., null, 256-274.
2. Peelaerts W., Bousset L., Van der Perren A., Moskalyuk A., Pulizzi R., Giugliano M., Van den Haute C., Melki R., Baekelandt V., (2015). α -Synuclein strains cause distinct synucleinopathies after local and systemic administration., Nature, 522, 340-4.
3. Mao Xiaobo., Ou Michael Tianhao., Karuppagounder Senthilkumar S., Kam Tae-In., Yin Xiling., Xiong Yulan., Ge Preston., Umanah George Essien., Brahmachari Saurav., Shin Joo-Ho., Kang Ho Chul., Zhang Jianmin., Xu Jinchong., Chen Rong., Park Hyejin., Andrabi Shaida A., Kang Sung Ung., Gonçalves Rafaella Araújo., Liang Yu., Zhang Shu., Qi Chen., Lam Sharon., Keiler James A., Tyson Joel., Kim Donghoon., Panicker Nikhil., Yun Seung Pil., Workman Creg J., Vignali Dario A A., Dawson Valina L., Ko Han Seok., Dawson Ted M., (2016). Pathological α -synuclein transmission initiated by binding lymphocyte-activation gene 3., Science, 353, undefined.
4. Sacino Amanda N., Brooks Mieu., Thomas Michael A., McKinney Alex B., Lee Sooyeon., Regenhardt Robert W., McGarvey Nicholas H., Ayers Jacob I., Notterpek Lucia., Borchelt David R., Golde Todd E., Giasson Benoit I., (2014). Intramuscular injection of α -synuclein induces CNS α -synuclein pathology and a rapid-onset motor phenotype in transgenic mice., Proc. Natl. Acad. Sci. U.S.A., 111, 10732-7.
5. Luk Kelvin C., Kehm Victoria., Carroll Jenna., Zhang Bin., O'Brien Patrick., Trojanowski John Q., Lee Virginia M-Y., (2012). Pathological α -synuclein transmission initiates Parkinson-like neurodegeneration in nontransgenic mice., Science, 338, 949-53.
6. Volpicelli-Daley Laura A., Luk Kelvin C., Lee Virginia M-Y., (2014). Addition of exogenous α -synuclein preformed fibrils to primary neuronal cultures to seed recruitment of endogenous α -synuclein to Lewy body and Lewy neurite-like aggregates., Nat Protoc, 9, 2135-46.
7. Bousset Luc., Pieri Laura., Ruiz-Arlandis Gemma., Gath Julia., Jensen Poul Henning., Habenstein Birgit., Madiona Karine., Olieric Vincent., Böckmann Anja., Meier Beat H., Melki Ronald., (2013). Structural and functional characterization of two alpha-synuclein strains., Nat Commun, 4, 2575.

3. The double labeling image in Figure 2 is very low power and difficult to appreciate the cellular localization of the α -Syn. The patches stained with MBP are unclear what they represent, fiber bundles?, they also need to show double label ubiquitin with cellular marker to understand better the cellular localization, the α -Syn aggregates in red only occasionally appear to co-localize with MAP2 and ubiquitin but nothing else, what is the cellular localization of other aggregates? Are they outside the cells?

R: Thank you for your meaningful suggestions that help us improve the quality of our paper. We have changed the double labeling high-power image in Fig. 2 in the revised manuscript (p. 33-35, lines 771-782). Myelin basic protein (MBP) is the second most abundant protein in the central nervous system myelin (PMID: 16794783) [1], the

anti-MBP antibody recognizes the epitopes in the myelin membrane, therefore the patches stained with MBP represent the myelin. Per your advice, we performed double-labeling immunofluorescence analysis for α -Syn aggregates and cellular markers. The results are shown in **Fig. 19** below. The immunofluorescence analysis exhibits that ubiquitin-immunoreactivity is localized inside neurons. Moreover, we also performed double-labeling immunofluorescence analysis using anti-p α -Syn antibody for α -Syn aggregates and anti-sodium potassium ATPase antibody (Abcam, ab76020) for cell membrane in order to confirm the cellular localization. The result shows that some aggregates are inside the cells (**Fig. 19 below**). Based on your suggestions, we have made some necessary modifications in the Result section of the revised manuscript (p. 5, 6, lines 109-130; p. 33-35, lines 771-782).

Fig. 19. Double immunofluorescence analysis of locus coeruleus from α -Syn PFFs mice for ubiquitin (green, A, C, D, F) with MAP-2 (red, B, C, E, F) and Na⁺/K⁺-ATPase (red, G and I) with p α -Syn (green, H and I). Co-immunolabeling is represented by signal in yellow. Cell nuclei were counterstained with Hoechst33258 (blue, F, I). [Scale bars, 500 μ m (A-C); 40 μ m (D-I)].

1. Boggs J M., (2006). Myelin basic protein: a multifunctional protein., Cell. Mol. Life Sci., 63, 1945-61.

4. Did they tried this experiment in the WT background? Why there is a need of low levels a-Syn tg to express, this means that their model requires templated seeding with homologous a-Syn (human) and not with mouse a-Syn as it has been shown in

the original studies of Luk et al. Is there expression of human α -Syn in the tg model in the periphery, they need to characterize. Is the effects dependent on the expression of the transgene in the autonomic nuclei.

R: Thank you for your thoughtful comments. We have conducted this experiment in C57BL/6 mice. We injected human α -Syn PFFs into bilateral SG and CG of C57BL/6 mice. In α -Syn PFFs-injected C57BL/6 mice, there is no pathological α -Syn and no phenotype can be induced. There is not any pathological change or phenotype in PBS-injected C57BL/6 mice (Please refer to the answer to Reviewer 2-Question 7). We agree with you, and consider that there might be a requirement of low levels of homogenous α -Syn transgene to induce the formation and transmission of α -Syn aggregates. A recent research published in *Neuron* this year (PMID: 31255487) [1] found that mouse α -Syn PFFs-injected mice showed a pathological α -Syn accumulation in the central nervous system at 3 months after injections into the pyloric stomach and upper duodenum of C57BL/6J mice. However, the human α -Syn PFFs-injected mice fail to elicit pathological α -Syn accumulation at the same time point. Besides, according to previous literature (PMID: 22508839) [2], intracerebral inoculation of human α -Syn PFFs initiates pathological and behavioral phenotypes in TgM83 mice which express a certain level of human α -Syn. This indicates that human α -Syn PFFs delivered exogenously can seed human α -Syn (A53T) into pathological forms in TgM83 mice. Exogenous injection of human α -Syn PFFs may not be able to initiate the α -Syn propagation and clinical phenotypes in C57BL/6 mice owing to the lack of human α -Syn in mice. This has prompted us to inject mouse α -Syn PFFs into the ganglia of WT mice in the future, and assess the pathological and behavioral changes in the mouse background.

To characterize the expression of human α -Syn in the TgM83 model in the peripheral and central nervous system, we had performed the immunohistochemical stainings in different tissues using anti-human α -Syn antibody (Abcam, syn211, ab80627). The immunohistochemistry results show extensive immunoreactivity to human α -Syn in both peripheral and central nuclei of uninjected TgM83^{+/-} mice (**Fig. 20** below). Therefore, we believe that the effects in our mouse model are dependent on the expression of the transgene in the autonomic nuclei. We suppose that there is also a necessity of low-level expression of human α -Syn to induce the formation and transmission of α -Syn aggregates in the autonomic nuclei. If the autonomic nuclei do not express human α -Syn transgene, the pathological α -Syn accumulation will not appear, which has been proven by the experimental results in C57BL/6 mice mentioned above.

Fig. 20. Representative immunohistochemical results of different segments from uninjected TgM83^{+/-} mice using anti-human α -Syn antibody. (A-F) The immunohistochemical images display segments including the stellate ganglia (A), celiac ganglia (B), gastric gland (C), colonic gland (D), dorsal motor nucleus of vagus (E), and the 3rd thoracic spinal cord (T3) (F). [Scale bars, 40 μ m].

1. Kim Sangjune., Kwon Seung-Hwan., Kam Tae-In., Panicker Nikhil., Karuppagounder Senthilkumar S., Lee Saebom., Lee Jun Hee., Kim Wonjoong Richard., Kook Minjee., Foss Catherine A., Shen Chentian., Lee Hojae., Kulkarni Subhash., Pasricha Pankaj J., Lee Gabsang., Pomper Martin G., Dawson Valina L., Dawson Ted M., Ko Han Seok., (2019). Transneuronal Propagation of Pathologic α -Synuclein from the Gut to the Brain Models Parkinson's Disease., *Neuron*, 103, 627-641.e7.
2. Luk Kelvin C., Kehm Victoria M., Zhang Bin., O'Brien Patrick., Trojanowski John Q., Lee Virginia M Y., (2012). Intracerebral inoculation of pathological α -synuclein initiates a rapidly progressive neurodegenerative α -synucleinopathy in mice., *J. Exp. Med.*, 209, 975-86.

5. It will be important to show that the spreading of the pathology is blocked by blocking the vagal nerve, otherwise the spreading mechanisms is unclear given that seeding is occurring in the background or a α -Syn tg model.

R: You made a very constructive suggestion. To evaluate whether pathologic α -Syn spreading via the vagus nerve in α -Syn PFFs-injected mice, we cut off the vagus nerve distributed near the gastrointestinal tract and injected α -Syn PFFs to celiac

ganglia 3 days later (**Fig. 21** blow). We found that there is no α -Syn inclusion in the dorsal nucleus of the vagus nerve in α -Syn PFFs-injected mice that accept truncal vagotomy even at 7 mpi. These mice still showed significant α -Syn pathology in other regions and nuclei as early as 2 mpi (**Fig. 21** below). We infer that the pathological α -Syn from the sympathetic ganglia could spread to heart, blood vessel, and gastrointestinal tract where it is taken in by the vagus nerve terminals and further spreads to the dorsal nucleus of vagus nerve transsynaptically retrogradely. The results of vagotomy indicate that exogenous α -Syn PFFs injected into stellate and celiac ganglia spread through both sympathetic pathway (via IML) and parasympathetic pathway (via the vagal nerve). Similar phenomenon has been demonstrated in a rat model of α -Syn PFFs spreading (PMID: 31254094) [1].

Therefore, we suppose that the presence of in pathological α -Syn in the dorsal nucleus of vagus nerve is a result of the pathological α -Syn retrogradely spreading from the vagus nerve terminals.

Following your suggestions, we have included **Fig. 21** below as a new Supplementary Fig. 3 (p. 7, lines 90-95) and made some necessary modifications in the revised manuscript (p. 16, 17; lines 349-358).

Fig. 21. (A and B) Photography (x40) of topographic anatomy of the vagal nerve before (A) and after truncal vagotomy (B) under the stereo-microscope. (C) Representative immunohistochemical results of medulla oblongata from pathological α -Syn PFF mice receiving truncal vagotomy. Pathologic α -Syn stained with anti-phospho- α -Syn (Ser 129) antibody. [Scale bar, 500 μ m]. (D) Schematic displaying the truncal vagotomy near the gastrointestinal tract and α -Syn PFFs injection to celiac ganglia.

1. Van Den Berge Nathalie., Ferreira Nelson., Gram Hjalte., Mikkelsen Trine Werenberg., Alstrup Aage Kristian Olsen., Casadei Nicolas., Tsung-Pin Pai., Riess Olaf., Nyengaard Jens Randel., Tamgüney Gültekin., Jensen Poul Henning., Borghammer Per., (2019). Evidence for bidirectional and trans-synaptic parasympathetic and sympathetic propagation of alpha-synuclein in rats., *Acta Neuropathol.*, 138, 535-550.

6. It would be important to compare to all the relevant controls to better understand to what extent the pathology and functional deficits are the results of synergy between the PFF and the M83 transgene or additive effects or unique to the PFF. The authors only show data on M83 PBS vs PFF, but would be important to compare to WT PBS vs PFF.

R: As mentioned above, we have conducted this experiment in C57BL/6 mice in the initial stage of the experiment. Please refer to the answer to your Question 4.

7. Need more detailed statistical analysis and to provide in the figure legends the n number per group. Only males were used, what is the explanation, why they didn't considered sex differences?

R: We greatly appreciate your careful observation. We used 20 mice in the bilaterally injected mice group and 10 mice in the unilaterally injected mice group for statistical analysis in each group. We have added the n number per group in the figure legends in the revised manuscript and supplement. We used both male and female mice in the pre-experiment and found no gender differences. We chose male mice in the formal study.

REVIEWERS' COMMENTS:

Reviewer #1 (Remarks to the Author):

I am satisfied with the responses to all my comments. The manuscript reads better and is more complete.

Ana Takakura

Reviewer #2 (Remarks to the Author):

Thank authors provided great efforts to revise in response my comments. They gave powerful data and contents which make the manuscript much better. I think the points raised in the previous round of review have been addressed except two more concerns about the revised figures as below.

Figure 2A, as authors considered that there is a correlation between non-motor symptoms and α -Syn pathology in different timelines, did you find out the correlation between them was positive or negative? If it was positive correlation, how about the Pearson correlation coefficient and P-value?

For figure 13E, please provide the scale bar on the TH staining.

Reviewer #3 (Remarks to the Author):

The authors had made a considerable effort reviewing the manuscript and nearly answering all the reviewers questions in the 41 pages provided in the rebuttal with additional tables, figures and references. The quality of the images have been improved, they have provided additional experimental evidence including assessing the effects of vagotomy and as requested provided additional double labeling studies and characterization of the PFF. They also provide a table 4 where they provide more clear explanation as to what is new in their model and experimental approach.

The only remaining concern is regarding the fact that the only way for the model to work is combining the M83 α -synuclein Tg with the PFF, it is surprising that the PFF alone injected in the ganglia of C57 wt mice does not work. No explanation is provided as to why the the PFF alone injected peripherally does not seed or spread aggregation. Maybe they can address this issue in the discussion.

REVIEWER #1

I am satisfied with the responses to all my comments. The manuscript reads better and is more complete.

Response to REVIEWER #1

Response (R): Thank you very much for your time and the positive comments. We greatly appreciate your constructive suggestions for improving our manuscript.

REVIEWER #2

Thank authors provided great efforts to revise in response my comments. They gave powerful data and contents which make the manuscript much better. I think the points raised in the previous round of review have been addressed except two more concerns about the revised figures as below.

Figure 2A, as authors considered that there is a correlation between non-motor symptoms and α -Syn pathology in different timelines, did you find out the correlation between them was positive or negative? If it was positive correlation, how about the Pearson correlation coefficient and P-value?

For figure 13E, please provide the scale bar on the TH staining.

Response to REVIEWER #2

R: Thank you for your time and constructive suggestions for improving our manuscript. We have carefully studied your comments and addressed them in the following parts.

1. Figure 2A, as authors considered that there is a correlation between non-motor symptoms and α -Syn pathology in different timelines, did you find out the correlation between them was positive or negative? If it was positive correlation, how about the Pearson correlation coefficient and P-value?

R: We really appreciate your careful observation and thoughtful suggestions. Following your suggestions, we analyzed the correlation between semi-quantitative grading of α -Syn pathology and non-motor symptoms using Spearman's rank correlation. As shown in **Fig.1** below, the Spearman's rank correlation coefficient between semi-quantitative grading of α -Syn pathology and orthostatic hypotension parameters is -0.8827 ($P=0.0444$). The Spearman's rank correlation coefficient between semi-quantitative grading of α -Syn pathology and defecation time is 0.9710 ($P=0.0111$). The Spearman's rank correlation coefficient between semi-quantitative grading of α -Syn pathology and the number of sweat spots is -0.9710 ($P=0.0111$). The Spearman's rank correlation coefficient between semi-quantitative grading of α -Syn pathology and the searching time in the buried food test is 0.8827 ($P= 0.0444$).

According to these results, we consider that there is a correlation between non-motor symptoms and α -Syn pathology in different timelines in the mouse model. The appearance of α -Syn aggregates parallels the onset of the non-motor symptoms. As the pathological deficits become more robust, the symptoms of the diseased mice are more severe. We have added **Fig. 1** below in the new Supplementary Fig. 2 in the revised Supplement and modified the Discussion and Methods section in the revised

manuscript, as can be seen in p.14, 26; lines 304-306, 558-560, 562.

Fig. 1. Semi-quantitative grading of α -Syn pathology versus the differences of the mean systolic pressure between the upright and supine position (a), defecation time of black feces (b), the number of sweat spots(c), and the searching time in the buried food test (d) of α -Syn PFFs mice. Grading of α -Syn pathology was performed as follows: 0, none; 1, slight; 2, moderate; 3, abundant; 4, severe.

2. For figure 13E, please provide the scale bar on the TH staining.

R: Thank you for your suggestions. We have added scale bars in the image of TH staining in Fig. 4k of the revised manuscript.

REVIEWER #3

The authors had made a considerable effort reviewing the manuscript and nearly answering all the reviewers questions in the 41 pages provided in the rebuttal with additional tables, figures and references. The quality of the images have been improved, they have provided additional experimental evidence including assessing the effects of vagotomy and as requested provided additional double labeling studies and characterization of the PFF. They also provide a table 4 where they provide more clear explanation as to what is new in their model and experimental approach.

The only remaining concern is regarding the fact that the only way for the model to work is combining the M83 a-synuclein Tg with the PFF, it is surprising that the PFF alone injected in the ganglia of C57 wt mice does not work. No explanation is provided as to why the the PFF alone injected peripherally does not seed or spread aggregation. Maybe they can address this issue in the discussion.

Response to REVIEWER #3

R: Many thanks for your precise comments and constructive suggestions for improving our manuscript.

We also injected human α -Syn PFFs into the stellate ganglia and celiac ganglia of C57BL/6 wildtype mice, and did not detect spreading of α -Syn. A recent research published in *Neuron* this year (PMID: 31255487) [1] found that mouse α -Syn PFFs-injected mice showed a pathological α -Syn accumulation in the central nervous system at 3 months after injections into the pyloric stomach and upper duodenum of C57BL/6J mice. However, the human α -Syn PFFs-injected mice fail to elicit pathological α -Syn accumulation at the same time point. Besides, according to previous literature (PMID: 22508839) [2], intracerebral inoculation of human α -Syn PFFs initiates pathological and behavioral phenotypes in TgM83 mice which express a certain level of human α -Syn. This indicates that human α -Syn PFFs delivered exogenously can seed human α -Syn (A53T) into pathological forms in TgM83 mice. Thus, there might be a requirement of homogenous α -Syn transgene to induce the formation and transmission of α -Syn aggregates. Exogenous injection of human α -Syn PFFs may not be able to initiate the α -Syn propagation and clinical phenotypes in C57BL/6 mice owing to the lack of human α -Syn in mice. This has prompted us to inject mouse α -Syn PFFs into the ganglia of WT mice in the future, and assess the pathological and behavioral changes in the mouse background. Thank you for your suggestion again. We have addressed this issue in the Discussion section in the revised manuscript, as can be seen in p.17, lines 364-367.

1. Kim Sangjune., Kwon Seung-Hwan., Kam Tae-In., Panicker Nikhil., Karuppagounder Senthilkumar S., Lee Saebom., Lee Jun Hee., Kim Wonjoong Richard., Kook Minjee., Foss Catherine A., Shen Chentian., Lee Hojae., Kulkarni Subhash., Pasricha Pankaj J., Lee Gabsang., Pomper Martin G., Dawson Valina L., Dawson Ted M., Ko Han Seok., (2019). Transneuronal Propagation of Pathologic α -Synuclein from the Gut to the Brain Models Parkinson's Disease., *Neuron*, 103, 627-641.e7.
2. Luk Kelvin C., Kehm Victoria M., Zhang Bin., O'Brien Patrick., Trojanowski John Q., Lee Virginia M Y., (2012). Intracerebral inoculation of pathological α -synuclein initiates a rapidly progressive neurodegenerative α -synucleinopathy in mice., *J. Exp. Med.*, 209, 975-86.